# Optimized photochemistry enables efficient analysis of dynamic RNA structuromes and interactomes in genetic and infectious diseases

Minjie Zhang[1,4], Kongpan Li[1,4], Jianhui Bai[1,4], Willem A. Velema[2], Chengqing Yu[1], Ryan van Damme[1], Wilson H. Lee[1], Maia L. Corpuz[1], Jian-Fu Chen[3] & Zhipeng Lu[1✉]

Direct determination of RNA structures and interactions in living cells is critical for understanding their functions in normal physiology and disease states. Here, we present PARIS2, a dramatically improved method for RNA duplex determination in vivo with >4000-fold higher efficiency than previous methods. PARIS2 captures ribosome binding sites on mRNAs, reporting translation status on a transcriptome scale. Applying PARIS2 to the U8 snoRNA mutated in the neurological disorder LCC, we discover a network of dynamic RNA structures and interactions which are destabilized by patient mutations. We report the first whole genome structure of enterovirus D68, an RNA virus that causes polio-like symptoms, revealing highly dynamic conformations altered by antiviral drugs and different pathogenic strains. We also discover a replication-associated asymmetry on the (+) and (−) strands of the viral genome. This study establishes a powerful technology for efficient interrogation of the RNA structurome and interactome in human diseases.

[1] Department of Pharmacology and Pharmaceutical Sciences, School of Pharmacy, University of Southern California, Los Angeles, CA, USA. [2] Institute for Molecules and Materials, Radboud University Nijmegen, Nijmegen, The Netherlands. [3] Center for Craniofacial Molecular Biology, University of Southern California, Los Angeles, CA, USA. [4] These authors contributed equally: Minjie Zhang, Kongpan Li, Jianhui Bai. ✉email: zhipengl@usc.edu

RNA structures and interactions play important roles in many cellular processes, ranging from carrying genetic information, to catalysis, regulation of gene expression, and beyond[1]. However, the vast majority of RNA molecules are too large and flexible for structure analysis using the physical methods, such as X-ray crystallography, NMR, and cryo-EM[2,3]. Base pair stacking is the dominant force in RNA structures and RNA–RNA interactions; therefore, direct determination of base pairs is a critical step toward decoding the structural basis of RNA-mediated regulation in cells. Recently, we and others developed approaches to determine RNA base pairs, based on the principle of crosslinking, proximity ligation, and high-throughput sequencing[4–9]. These methods, including PARIS (psoralen analysis of RNA interactions and structures), SPLASH, LIGR-seq, and COMRADES, allowed direct analysis of RNA duplexes at the transcriptome level, achieving single-molecule accuracy and near base pair resolution. However, despite over 50 years of research, our understanding of the physical, chemical, and enzymatic properties of "crosslink-ligation" methods remain limited, hindering their applications to various biological systems.

We have now systematically investigated the basic physics and chemistry of the crosslink-ligation principle; and developed next generation of the PARIS method (PARIS2). In particular, we report amotosalen as a more efficient crosslinker compared to the commonly used psoralen AMT. We discover that crosslinking increases RNA hydrophobicity, rendering it unextractable using the classical AGPC (acid guanidine thiocyanate phenol chloroform) aqueous–organic phase separation method (commercially known as TRIzol, etc.) or silica-based solid-phase extraction methods[9,10]. We invent a generally applicable method, TNA (total nucleic acid extraction), to purify crosslinked RNA, enabling targeted analysis of RNAs with antisense enrichment. Given the low efficiency crosslinking, several methods have been developed to enrich crosslinked fragments, including native-denatured two-dimension (ND2D) gel, biotin-tagging and RNase R treatment; however, these approaches are often expensive and inefficient[2]. We develop a denatured–denatured 2D (DD2D) gel system to isolate pure crosslinked RNA without the need for tagging the crosslinker. We also introduce chemical and enzymatic approaches to prevent and bypass photochemical damages to RNA, a fundamental problem in RNA research. Together, these optimizations in PARIS2 resulted in >4000-fold increased efficiency, and importantly, the deep mechanistic insights into photochemistry, RNA chemistry, and enzymology for individual improvements are also broadly applicable in RNA studies.

We demonstrate the power of the PARIS2 method in three applications. Using enriched mRNAs, we discover that crosslinked RNA fragments can report the translation status of mRNAs on a global scale. We discover a network of dynamic RNA structures and interactions in the U8 snoRNA involved in ribosomal RNA (rRNA) processing. Mutations in U8 that cause the neurological disorder leukoencephalopathy with calcifications and cysts (LCC) disrupt this RNA network. We applied PARIS2 to determine the dynamic genome architecture of enterovirus EV-D68, a reemerging RNA viral pathogen associated with severe neurological symptoms, and discover novel structure conformations. We found that mutations in one particular alternative conformation of the EV-D68 IRES reduces translation efficiency, suggesting functional significance of structure dynamics. The PARIS2 method will enable more rapid and facile analysis of structural basis of RNA functions in various biological systems and human diseases.

## Results

### Overview of the PARIS2 strategy and major improvements.

The crosslinking and proximity ligation-based principle for RNA secondary structure and interaction analysis relies on the successful completion of multiple reaction and extraction steps (Fig. 1a). The process starts with psoralen crosslinking in live cells, followed by RNA extraction and fragmentation, isolation of crosslinked from non-crosslinked, proximity ligation, crosslink reversal, adapter ligation, reverse transcription, and finally cDNA amplification. In this study, we performed a systematic analysis of each step and make improvements based on newly discovered physical, chemical, and enzymatic properties of RNA reactions and extractions (summarized in Fig. 1b). The improvements include (1) high solubility and high efficiency crosslinker amotosalen, (2) complete extraction of crosslinked RNA, (3) simplified RNA fragmentation using RNase III, (4) DD2D gel selection of crosslinked RNA, (5) optimized adapter ligation, and (6) prevention and bypass of photochemical RNA damage. Together these changes lead to >4000-fold improvement in the efficiency for PARIS2 (Supplementary Tables 1–3). Major optimizations are presented below, whereas additional mechanistic studies and exhaustive screens are described in Supplementary Notes.

### Highly soluble psoralen amotosalen increases RNA crosslinking efficiency.

The commonly used psoralen AMT is only soluble at 1 mg/ml in water, limiting crosslink efficiency[11] (Fig. 2a). We designed a three-step method to synthesize amotosalen, another psoralen derivative[12] (Fig. 2a), and found it soluble at 230 mg/ml in water and efficiently crosslink oligos in vitro (Supplementary Figs. 1 and 2). We discovered that crosslinking repartitions large RNA from the aqueous phase to the interphase during standard AGPC (TRIzol) extraction (Fig. 2b). The migration of crosslinked RNAs from aqueous to the interphase serves as an indicator of crosslinking efficiency. Both total yield and 18S + 28S percentage in the aqueous phase were reduced with higher psoralen concentrations, suggesting higher crosslink efficiency (Fig. 2b, c). Purified crosslinked fragments from psoralen-crosslinked cells increased from 0.67 to 4.65%, ~7-fold, after a 10-fold increase in crosslinker concentration (AMT 0.5 vs. amotosalen 5 mg/ml; Fig. 2d, Supplementary Fig. 2c, d, and Supplementary Tables 1 and 2). The RNA duplexes captured by AMT and amotosalen (both at 0.5 and 5 mg/ml) are similar at

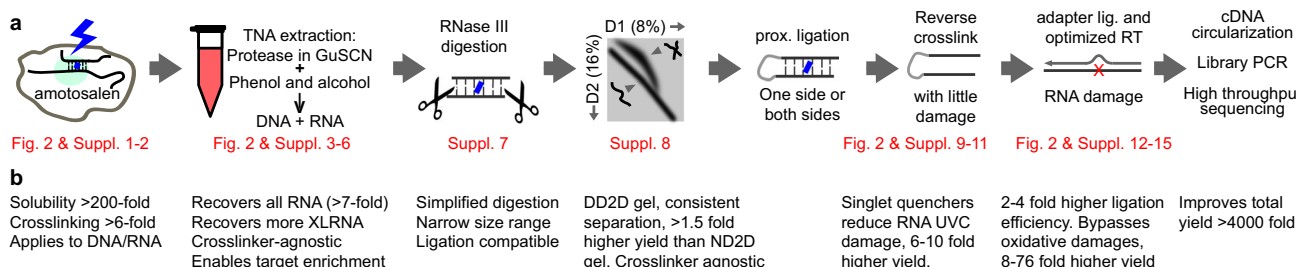

**Fig. 1 PARIS2 outline and summary of improvements. a** Outline of the PARIS method and major improvements. Details of the improvements are presented in subsequent figures and tables. **b** Notes on the major advantages of the new method and fold improvement for each step.

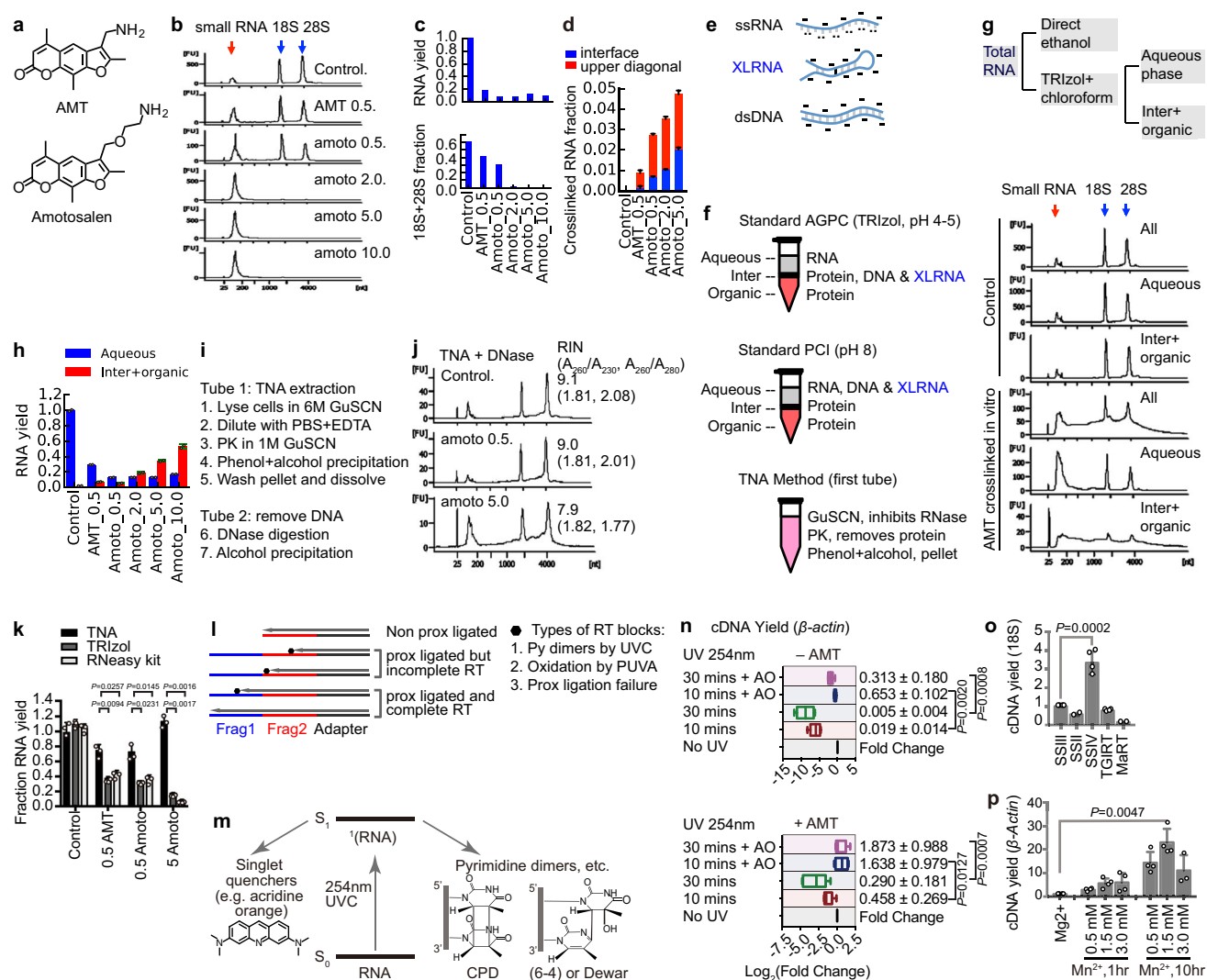

**Fig. 2 Systematic optimization of PARIS2. a** Structures of AMT and amotosalen. **b** Higher concentrations of crosslinkers increase crosslinking efficiency. Psoralen crosslinked HEK293 cells RNA was extracted using the standard AGPC (TRIzol) method. Small RNA: RNA in the range of 50–300 nt, including tRNA, snRNA, 5S and 5.8S rRNAs etc. **c** Total RNA yield (upper panel) and 18S + 28S percentage (lower panel) from panel (**b**). **d** Fraction of psoralen crosslinked RNA from DD2D gels. Data are mean ± s.d.; $n = 2$ biological replicates. **e** Charge and hydrogen bonding of RNA and DNA molecules in standard AGPC (TRIzol) purification. The '-' indicates negative charges. The '..' indicates exposed bases involved in hydrogen bonding. **f** Phase partition of DNA, RNA and protein in AGPC and PCI methods, and comparison to TNA. **g** Repartition of crosslinked RNA to the interphase in Trizol extraction. Purified total RNA was treated with or without 365 nm UV plus AMT, and then either directly precipitated using ethanol, or extracted from the 2 phases of TRIzol-chloroform. **h** Recovery of crosslinked RNA from the aqueous and inter+organic phases using the TNA method. Data are mean ± s.d.; n = 2 biological replicates. **i** TNA method outline. **j** Recovery of inact RNA from psoralen crosslinked HEK293 cells using TNA method, after DNase treatment. RNA integrity numbers (RIN) and two indicators of RNA quality, A260/A230 and A260/A280, were listed. **k** Quantity of RNA isolated from control and AMT/amotosalen crosslinked cells. Data are mean ± s.d.; $n = 3$ biological replicates. two-tailed, unpaired $t$-test. **l** Effects of UV damages and failed proximity ligations (non prox ligated) on the yield of gapped reads. **m** UVC excited RNA form various products, such as pyrimidine dimers, hydrates and strand breaks. Alternatively, the energy can be transferred to singlet quenchers like acridine orange, leaving RNA intact. S0 and S1: ground state and excited singlet state. **n** cDNA yield from RNA irradiated with 254 nm UV, with or without AO protection, normalized to non-photo-reversal sample. Top: Non-crosslinked. Bottom: AMT crosslinked. Box plots show center line as median, box limits as upper and lower quartiles, whiskers as minimum to maximum values. 10 min + AO: $n = 4$; others: $n = 7$, biological replicates. two-tailed, unpaired $t$-test. **o** cDNA yield obtained in RT-qPCR experiments for PUVA damaged 18S-rRNA, normalized to SSIII. MaRT: Marathon RT. Data are mean ± s.d.; SSIII/SSIV/TGIRT: $n = 4$; SSII/MaRT: $n = 2$, biological replicates. two-tailed, unpaired $t$-test. **p** cDNA yield for β-Actin using SSIV in different reaction buffers and different incubation time, normalized to a standard Mg2+ condition. Data are mean ± s.d.; $n = 4$ biological replicates. two-tailed, unpaired $t$-test.

equivalent sequencing coverage, both on a global scale and in specific RNAs (Supplementary Fig. 2e–m). Therefore, we identified the highly soluble amotosalen as a more efficient crosslinker for RNA structure studies.

**Phase partition and extraction of crosslinked RNA.** In the classical AGPC method for RNA extraction, the mixture of guanidine thiocyanate (GuSCN), phenol, and chloroform forms two phases[10]. RNA partitions to the aqueous phase at pH <5, proteins partition to the inter and organic, while DNA partitions to the interphase (Fig. 2e, f and Supplementary Note 1). In contrast, the standard PCI (phenol, chloroform, and isoamyl alcohol) method for DNA extraction employs higher pH to bring both DNA and RNA to the aqueous phase. While applying AGPC

(e.g., TRIzol), we noticed that crosslinking greatly reduced RNA yield[9] (Supplementary Fig. 3a). We used proteinase K (PK) and RNase digestion in lysate to recover crosslinked RNA; however, the yield was low, and fragmentation prior to purification makes it difficult to enrich specific RNAs with antisense oligos[9]. We speculated that crosslinked RNA may be more hydrophobic and partitioned to the interphase. To test this possibility, we crosslinked pure total RNA with AMT and directly precipitated RNA with ethanol (Fig. 2g, h). Alternatively, we used the AGPC method (TRIzol) to separate the three phases and then precipitated RNA from each phase. Crosslinking induced a broad smear that spans beyond the largest 28S peak. While direct ethanol precipitation recovered all RNA, the TRIzol–chloroform aqueous phase contains only RNA between 50 and 300 nt (e.g., tRNAs, snRNAs, and snoRNAs) and sharp non-crosslinked 18S/28S (Fig. 2g). Adding formamide to the TRIzol–chloroform mixture shifted crosslinked RNA to the aqueous phase (Supplementary Fig. 3b). These results suggested that crosslinking increased RNA hydrophobicity, which reduced yield from the classical AGPC method. In all previous studies, the inefficient recovery of crosslinked RNA likely has resulted in significant bias because larger and heavier crosslinked RNA molecules are selectively lost[5–7].

To recover total RNA after crosslinking, we developed a method, TNA (Fig. 2i and Supplementary Fig. 3c–g). Briefly, cells are first lysed in 6 M GuSCN to completely inhibits nucleases. The lysate is diluted to reduce GuSCN concentration, treated with EDTA to chelate divalent cations, and digested with PK to remove proteins. Adding phenol and alcohol precipitates total nucleic acids without protein contaminants. Subsequent DNase treatment affords pure crosslinked RNA (Fig. 2j). Carmustine and chlorambucil, two chemotherapy drugs that crosslink nucleic acids, also increased RNA hydrophobicity, and crosslinked RNA was successfully recovered using TNA (Supplementary Figs. 4 and 5). TNA outperforms both AGPC and solid-phase methods by at least sixfold (Fig. 2k, Supplementary Fig. 6, and Supplementary Tables 1–3). These results showed that crosslink-induced hydrophobicity is a general property of RNA, and TNA is generally applicable to crosslinking studies. The intact purified RNA makes it possible to efficiently enrich crosslinked RNA using antisense oligos.

**Efficient isolation of crosslinked RNA using a DD2D gel system.** To obtain short crosslinked RNA fragments, we developed a simplified one-step RNase III protocol that takes advantage of the digestion kinetics (Supplementary Fig. 7 and Supplementary Note 2). Given the low efficiency of psoralens, crosslinked fragments need to be enriched for sequencing. Previously reported biotin-conjugated psoralens require custom synthesis and are contaminated with psoralen monoadducts, which are more abundant than crosslinks[5,7]. RNase R depletion of non-crosslinked RNA is also impeded by monoadducts[6]. Here, we develop a DD2D gel system that takes advantage of the differential migration of crosslinked RNA vs. non-crosslinked at different gel concentrations, during electrophoresis to isolate pure crosslinked RNA without monoadduct contamination (Supplementary Fig. 8 and Supplementary Note 3). The DD2D method is more efficient than our previous method ND2D (1.5-fold higher yield), outperforms biotin-tagging and RNase R enrichment, and is generally applicable to different crosslinkers[9,13] (Supplementary Figs. 4, 5, and 8).

**Prevention of RNA against UVC damages and bypass of PUVA damages.** Photochemical crosslinking (psoralen + UVA 365 nm) and reversal (UVC, 254 nm) enable in vivo analysis of RNA duplexes, but also cause many types of damage. Together with the low efficiency proximity ligation, the damages block reverse transcription and reduce both the total cDNA yield and percentage of gapped reads (Fig. 2l). UVC irradiation induces pyrimidine dimers and other damages via the singlet excited state, even after very short exposure[14,15] (Fig. 2m). Singlet quenchers have been shown to block UVC-induced DNA, but not RNA damages[16,17]. To prevent, repair or bypass UVC-induced RNA damages, we systematically screened a variety of conditions (Supplementary Figs. 9–11 and Supplementary Note 4). Superscript IV (SSIV) reverse transcriptase outperforms other enzymes on UVC-damaged RNA, increasing yield by sevenfold over SSIII (Supplementary Fig. 9). Singlet quenchers acridine orange (AO) and ethidium bromide (EB) at high concentrations can protect normal and psoralen-crosslinked RNA from UVC irradiation. AO effectively protects non-crosslinked RNA even after 30 min UVC irradiation (at 4 mW per cm$^2$), after which 30% RNA remain intact, vs. 0.5% in the absence of AO (Fig. 2n, upper panel). AO also protects psoralen crosslinked from UVC damage (Fig. 2n, lower panel), without blocking reversal (Supplementary Fig. 10i–l), making it possible to apply them in PARIS-like experiments. Together, we demonstrate that UVC-induced RNA damage can be prevented by high concentrations of singlet quenchers. After proximity ligation and reversal of crosslinks, RNA samples are then ligated with adapters for reverse transcription and library preparation (see optimizations in Supplementary Fig. 12).

In addition to crosslinking pyrimidines, photosensitized psoralens also induce oxidative damage, primarily affecting guanines through direct electron transfer and excitation of oxygen[18] (Supplementary Fig. 13). We systematically screened conditions to prevent, repair, or bypass these oxidative damages (Supplementary Figs. 13–15 and Supplementary Note 5). RNA damage impedes reverse transcription by trapping the enzyme in an inactive state[19,20]. We reasoned that conditions that promote enzyme conformation dynamics and longer incubation time may overcome such barriers. Indeed, SSIV, Mn$^{2+}$, and longer incubation time dramatically increased cDNA yield both alone or in combination, in both primer extension assays and qRT-PCRs (Fig. 2o, p and Supplementary Fig. 15). Together, these conditions for SSIV improved the bypass of PUVA-induced damages by 8–70-folds over SSIII (Supplementary Tables 1–3).

After optimizing all individual steps, we tested their performance in PARIS2 workflow. Starting from the same number of cells, the 5 mg/ml amotosalen crosslinking, TNA extraction and DD2D gel isolation improved the yield of crosslinked RNA fragments by >60-fold over the standard AMT-TRIzol-ND2D protocol. Starting from the same amount of crosslinked RNA fragments (after DD2D gel step), the DNA library yield is improved ~76-fold (Fig. 1, and Supplementary Tables 1 and 2). Together the improvements resulted in a total of >4000-fold increase in efficiency. For oligo(dT)-enriched RNA, we were able to model structures of abundant mRNAs even with only ~1 M gapped reads for all mRNAs (Supplementary Fig. 16).

**PARIS2 enables profiling of ribosome SSU binding across the transcriptome.** During translation, mRNAs directly contact the 18S rRNA in the small subunit (SSU)[21] (Fig. 3a, b). We reasoned that psoralen crosslinking of mRNA to rRNA may allow direct analysis of translation. Mapping PARIS2 data to the engineered genome references with single-copy rRNAs (Supplementary Figs. 17 and 18), we found that mRNAs are specifically crosslinked to 18S helix 18 and 26 (h18 and h26), both of which are in the ribosome mRNA channel (Fig. 3a–c and Supplementary Fig. 19a, b). On the mRNA side, the strongest binding is on the

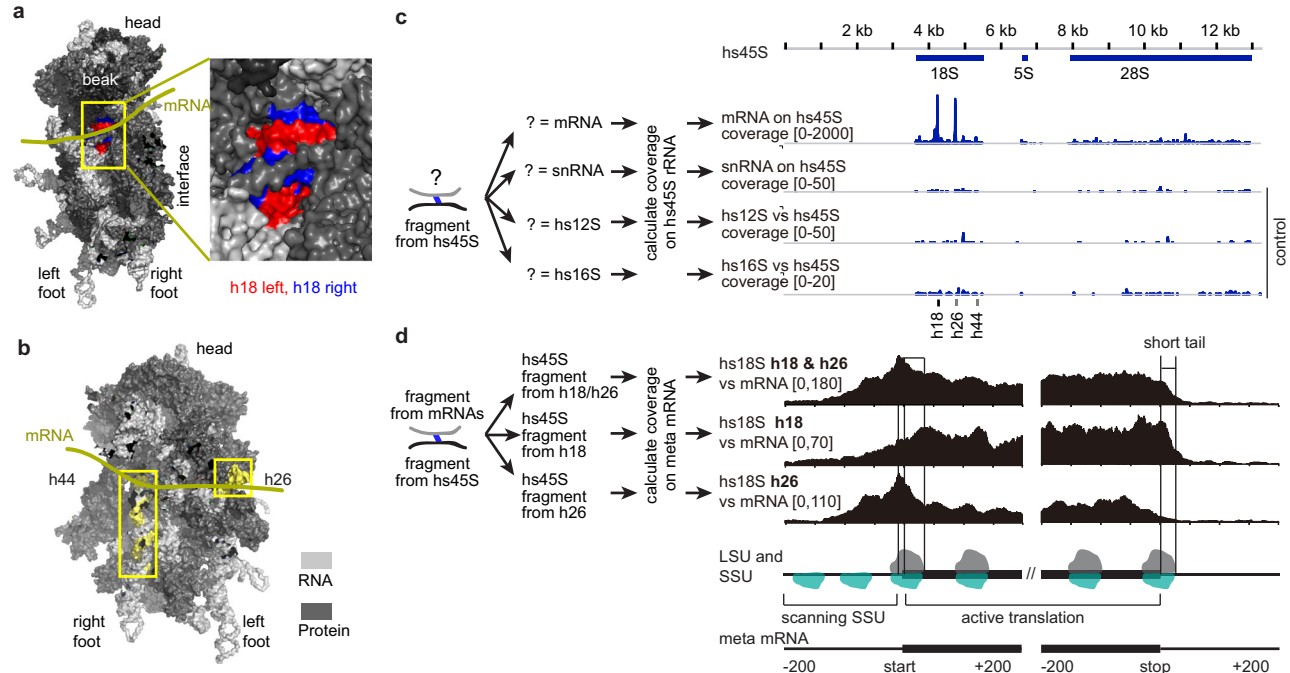

**Fig. 3 PARIS2 enables global profiling of SSU ribosome binding sites. a** The 40S ribosome small subunit (SSU), showing the beak side view, highlighting the 18S h18 helix exposed in the mRNA channel. rRNA in light gray, RPS in dark gray, except h18 left arm: 595–620, right arm: 621–641. **b** Rotated 40S, showing h26 and h44 in yellow. **c** The binding sites of mRNAs, snRNAs, and mitochondrial rRNAs (12S and 16S) on the 45S unit based on HEK293 mRNA PARIS2 data. **d** The binding sites of h18 and h26 on the meta mRNA. The tapering off signal on the 5′UTR is likely due to its limited length. The sharp drop off after the stop codon (20–30 nt short tail) is due to random RNA fragment length.

coding sequence, followed by the 5′UTR, and the 3′UTR (Fig. 3d). The highest peak is right next to the start codon. Interestingly, h26 binding site precedes that of h18, consistent with their locations in the mRNA channel. The mRNA–rRNA crosslinking could be a result of dynamic flipping of the h18 and h26 bases that transiently pair with mRNAs[21]. The binding in the 5′UTR but not 3′UTR may represent the scanning phase of translation initiation, which has been previously captured in translation complex profiling[22,23]. Similar patterns of rRNA–mRNA interactions were observed in individual mRNAs and in mouse brain oligo(dT)-enriched RNAs, HEK293 total RNAs, and mouse ES cell total RNAs, confirming the specificity of these interactions (Supplementary Fig. 19c–h). Comparison of mRNA–rRNA interactions in PARIS2 with ribosome profiling data from the same cell type[24] revealed high concordance on a global level, as well as patterns on meta-mRNAs and individual mRNAs (Supplementary Fig. 19i–m). Together, we demonstrate PARIS2 as a powerful alternative method for direct analysis of mRNA translation. PARIS2 does not replace ribosome profiling, due to its lower efficiency (~10% reads are gapped); however, the ability to capture translation status is a bonus during the analysis of mRNA structures and interactions.

**PARIS2 reveals a dynamic RNA structure and interaction network in ribosome biogenesis and LCC.** LCC is a neurodegenerative disorder caused by mutations in the snoRNA gene U8 (*SNORD118*)[25–27]. We recently discovered that U8 specifically binds the 3′ end of the 28S rRNA[9]. To understand the structural basis of U8 in ribosome biogenesis and LCC etiology, we used antisense oligos to enrich snoRNAs from crosslinked human HEK293 cells and mouse brains. PARIS2 data revealed five alternative conformations of U8 in a dynamic network with U13 snoRNA and 18S/28S rRNAs (Fig. 4a, b and Supplementary Fig. 20). The U8:U13 interaction suggests a molecular bridge

coordinating the biogenesis of rRNA subunits 18S and 28S (Supplementary Fig. 20m, n). The mutually exclusive base pairings among the interactions suggest dynamic rearrangement of the duplexes during rRNA processing. Although the U8 binding site is ~500 nt away from 28S 3′ end, it is only 18 nt away in physical space, and among the most stable in all putative interactions with the 45S, consistent with its role in the 28S 3′ end processing (Supplementary Fig. 21a–e). U8 depletion in human cells reduced 28S level relative to 18S (Supplementary Fig. 21f, g), suggesting a direct role in 28S processing, consistent with early studies in *Xenopus*[28].

To determine the molecular effects of LCC mutations, we systematically profiled minimal free energy (MFE) changes in ten duplexes in the U8 network (Fig. 4c). Among 32 distinct single-nucleotide mutations (deletion, duplication, insertion, and substitution) mapped to U8, 29 altered the stability of at least one duplex. Out of the 320 combinations (10 duplexes and 32 mutations), 12 changed stability by at least 5 kcal/mol (11 destabilizing and 1 stabilizing). Most changes cluster on the 5′ end domain involved in the five dynamic conformations, including the interaction with 28S. The U8 homodimer interaction was most destabilized since each mutation in the dimerization sequence affects both arms (Fig. 4c and Supplementary Fig. 21h). Using synthetic RNA oligos for the 22 nt of the 5′ end of U8 that forms multiple alternative conformations, we confirmed that a single 3G > A mutation found in LCC patients dramatically reduced dimer stability (Supplementary Fig. 21h, i). Together, these results raised the possibility that LCC mutations affect ribosome biogenesis by disrupting the network of snoRNA structures and interactions.

**PARIS2 determines the genome structure of the enterovirus EV-D68.** The genomes of RNA viruses carry the genetic information, and at the same time fold into complex structures to

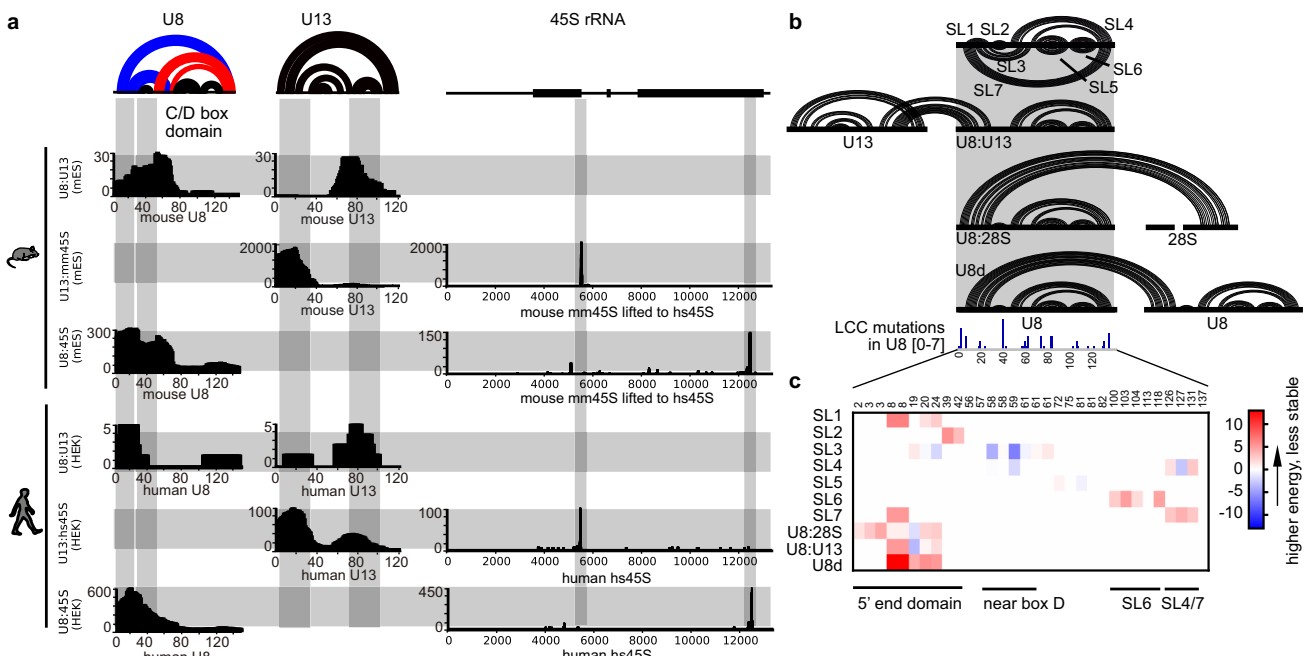

**Fig. 4 Discovery of a snoRNA–rRNA interaction network in rRNA processing and LCC. a** Intermolecular interactions among U8, U13, 18S, and 28S based on PARIS2 data from human HEK and mouse brain. Two intramolecular U8 alternative conformations are labeled in red and blue colors. Each row illustrates interactions between one pair of RNAs. Each column illustrates regions in one RNA that participate in interactions. **b** Mapping LCC mutations onto the U8 interaction network. Stemloops (SL) are defined for all U8 conformations and their interactions. U8d U8 dimer. All mutations from LCC patients were summarized in the track, where the coverage [0–7] indicates counts of mutations. **c** MFE changes for all U8 related duplexes due to LCC-causing mutations, calculated using PARIS2-derived models and RNAcofold. Red: that mutations increased the MFE and lowered stability of the duplex.

regulate multiple steps of their infection life cycles. However, the direct analysis of viral genome structures and interactions in cells remains challenging. We applied PARIS2 to EV-D68, a ~7300 nt positive strand RNA virus in the *Picornaviridae* family, whose recent global outbreaks have been associated with severe respiratory symptoms and acute flaccid paralysis (AFP), which resembles poliomyelitis[29] (Fig. 5a). To understand the structural dynamics of EV-D68 genomes in evolution, antiviral treatments, and different host cell types, we used two strains, the prototype VR1197 (F02-3607 Corn), and neurotropic, AFP-causing US/MO/14-18947 (US47; Fig. 5b). The cancer cell line HeLa and the neuronal SH-SY5Y were infected and treated with two antiviral drugs, Rupintrivir (AG7088)[30], which inhibits the viral protease 3C^pro and subsequently blocks replication, and geldanamycin (GA)[31], which blocks assembly of capsids into virion particles (Fig. 5c, d and Supplementary Fig. 22a–f). The efficient enrichment of viral RNAs in PARIS2 (Supplementary Fig. 22g–h) allows us to build reliable duplex groups (DGs) and secondary structure models, and determine structure dynamics in various genetic backgrounds and experimental conditions[9,32] (Fig. 5e, and Supplementary Figs. 23 and 24a, b).

PARIS2 determined viral genome structures are highly consistent in the two cell lines (Fig. 5f and Supplementary Fig. 24c), suggesting little effect of cell type on the overall structure. The two strains are more different from each other, where the conserved structures are mostly local (Fig. 5g, h and Supplementary Fig. 24d). Upon antiviral treatments, the overall structures remain mostly conserved, with minor differences (Supplementary Fig. 24e, f, see analysis of differences later), further supporting the robustness of the method and the stability of structures in vivo. PARIS2 confirmed previously reported secondary structure, including the 5′CL (cloverleaf), IRES (internal ribosome entry site), and the 2C-CRE (cis-acting replication element), which play critical roles in replication and translation[33,34] (Supplementary Figs. 25–27). Integration of PARIS2

and phylogenetic conservation analysis revealed a subset of highly conserved duplexes that are potentially functional in the EV-D68 genomes (Supplementary Figs. 26 and 27). PARIS2 directly captures RNA duplexes at a single-molecule level, therefore enabling the direct discovery of alternative conformations (Fig. 5e, i, j and Supplementary Fig. 28). Alternative and dynamic conformations are present along the entire length of the viral genomes, with ~80% DGs (cov ≥ 0.01) and ~30% DGs (cov ≥ 0.05) involved in alternative conformations. This result suggests that building a single structure conformation for viral genomes, which is a common practice in the field, would lead to erroneous models[35].

The (+) and (−) strands serve as templates for copying each other in viral genome replication. The transient base pairings between the two strands allow us to capture both strands for structure analysis, using antisense oligos targeting the (+) strand (Fig. 5k and Supplementary Fig. 29). In theory, complementary sequences should form structures that are close to mirror images of each other (Fig. 5l). However, we found that the (+) strands has more long-range structures than the (−) strand (Fig. 5m–p and Supplementary Fig. 29). This is likely due to their different abundance, with the (+) outnumbering the (−) by ~70-fold, leading to most (−) RNA in the double-stranded state with (+) RNA[36] (Supplementary Fig. 29d). AG treatment also consistently reduces long-range structures (Fig. 5m), likely reflecting preferential inhibition of (+) strand synthesis and a higher fraction of (+) RNA engaged in translation[37,38]. Interestingly, the 3′ end of the (−) RNA, encompassing the last 600 nt (first 600 nt on the (+) strand, including the 5′CL and IRES), is significantly less structured than the rest of the genome (Fig. 5p and Supplementary Fig. 29g). This is likely due to the frequent priming of (+) RNA synthesis that sequesters the 3′ end of (−) RNA. Together, these studies report the first global structure analysis of the (−) strand of (+) RNA viruses, revealing replication-associated differences in the long-range structures and the 3′ end of the (−) strand.

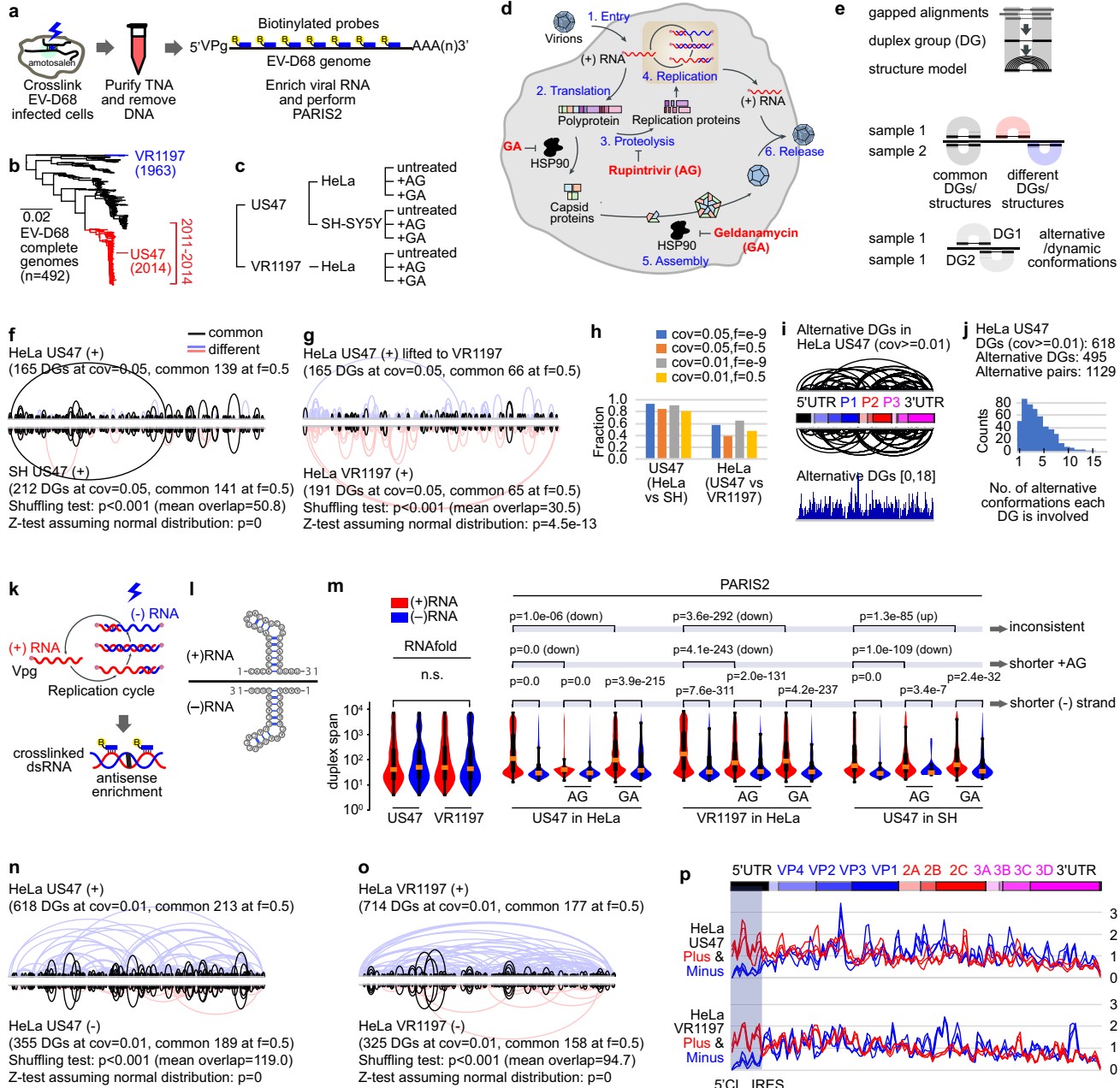

**Dynamic and long-range structures in the EV-D68 5′UTR.** The 5′UTR contains 5′CL and IRES, two elements critical for genome replication and translation; however, their in vivo conformations remain unknown. Using PARIS2, we identified all previously predicted structures, and discovered new alternative conformations in both strains (Fig. 6a). These alternative conformations are highly abundant (read numbers in parentheses) and supported by multiple sequence alignments in all clinical EV-D68 isolates, strongly indicating functional relevance (Supplementary Fig. 31). Disruption of either domain IV or its alternative conformation, a3, using two different sets of mutations, caused significant reduction in translation efficiency based on a bicistronic reporter assay (Supplementary Fig. 32). The overlap of alternative conformations poses significant challenges in the design of compensatory rescue mutations. Compensatory mutations in a3 did not rescue the IRES activity, likely due to the disruption of the overlapping domain IV. Combining mutations in both domain IV and a3 caused more significant reduction, suggesting additive effects between domain IV and a3 in

IRES activity. Together, these results provided further support for the functional relevance of alternative conformations.

In addition to local structures, the 5′UTR forms different long-range structures with the rest of the genome of the two strains (Fig. 6b and Supplementary Fig. 23e). The left-side anchors of the long-range structures are located in the linkers (L1–L3) among the first four domains, suggests a hierarchical folding process (Fig. 6c). The strongest right-side anchors are located, in US47, near the start codon and 2C-CRE, and in VR1197, near the stop codon (Fig. 6b, d–g, and Supplementary Fig. 23e and 33a–i). To quantify the structure dynamics in evolution, we computed the fraction of alignments supporting these conformations and found that the L1-CRE and L2L3-start in US47, and L1L2-3D-stop structure in VR1197 are mutually exclusive (Fig. 6h). Furthermore, the L2L3-start structures in US47 are lost upon AG treatment in HeLa cells, but not SH-SY5Y cells, indicating cell-type and strain-specific functions of this long-range structure (Fig. 6h and Supplementary Fig. 33a–f). The thermodynamic stability of the three conformations

**Fig. 5 PARIS2 reveals dynamic structuromes of EV-D68 viral RNAs in cells. a** Schematic diagram of the experimental strategy. Virus-infected HeLa cells were crosslinked. RNA was extracted using the TNA method and the viral genome RNA was enriched using biotinylated oligos. VPg viral genome-linked protein. **b** Two strains of EV-D68 were chosen: VR1197 (isolated in 1963) and US47 (US/MO/14-18947, isolated in 2014). **c** Experimental conditions. HeLa and SH-SY5Y cells were infected with the two strains and treated with two inhibitors, AG and GA. **d** The enterovirus life cycle and inhibitors mechanisms of action. **e** Analysis strategies. Gapped alignments were first assembled into duplex groups using CRSSANT. DGs in two samples are compared to identify common and different ones. DGs within the same sample are compared to each other to identify alternative/dynamic conformations. **f** US47 genome structure is highly reproducible in HeLa and SH cell lines based on shuffling test (1000 times). cov relative coverage, f fraction overlap, mean overlap average number of overlaps from the 1000 shuffles. **g** Local structures are conserved between US47 and VR1197 strains in HeLa cells. Data in the US47 strain were lifted to the VR1197 coordinates. **h** Fractions of overlaps calculated at different relative coverage (cov) and fraction overlaps of the two arms (**f**). **i, j** PARIS2 detects extensive alternative conformations along the entire length of US47 genome. Relative coverage at least 0.01 (cov ≥ 0.01) was set as cutoff for duplex groups. Arcs on the top and bottom represent DGs that are involved in alternative conformations with each other. The alternative DGs track indicates number of alternative conformations each region is involved in, which is then summarized in a histogram (**j**). **k** Diagram showing the recovery of crosslinked dsRNA intermediates during replication, using antisense probes targeting the plus (+) strand. **l** Diagram showing the theoretical structures on the (+) and (−) strands, which are mirror images of each other (with the exception of G–U pairs, where the counterpart A–C pairing is less stable). **m** Comparison of duplex span on the two strands in all experimental conditions. Duplex span for RNAfold-predicted structures (default parameters) is calculated as the linear distance between base pairs. Duplex span for PARIS2-derived structures is the linear distance between the middle points of the two arms. All primary gap1 alignments were used for calculation. Distances were log-transformed before plotting the violin and box plots. In the box plots, wiskers represent the max and min. The top and bottom of the box represent the first and third quartiles. The red bar is the median. Here are the numbers of samples for each of the 22 violin + box plots. For RNAfold: n = 2223, 2248, 2287, 2290. For US in HeLa: 235,976, 3958, 150,676, 5538, 238,185, 2925. For VR in HeLa, 159,270, 2874, 89,940, 1687, 124,774, 1935. For US in SH: n = 127,123, 1744, 6167, 46, 74,388, 352. P values were calculated using the Mann–Whitney U test. **n, o** Comparison of DGs on the (+) and (−) strands in US47 (**n**) and VR1197 (**o**) strains. **p** Comparison of structure density on the (+) and (−) strands in the two strains. The three samples for each strain were colored red or blue, for the (+) and (−) strands, respectively. The gray shadowed area highlights the biggest difference in the 5′CL and IRES structures between the two strands.

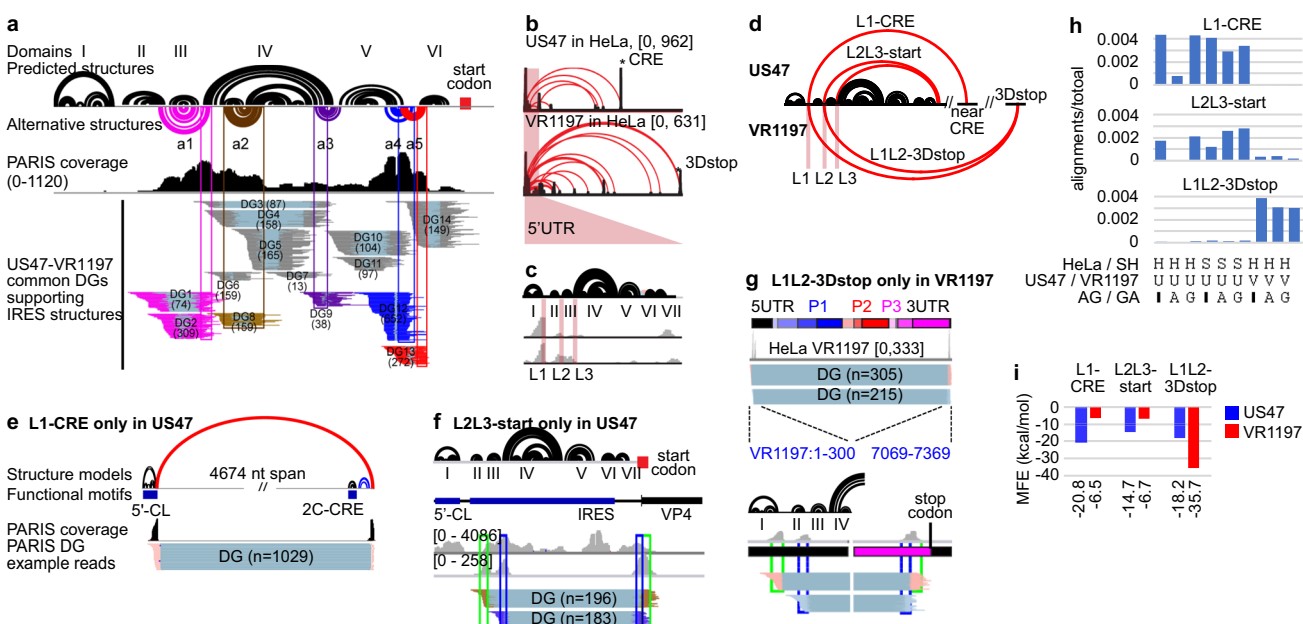

**Fig. 6 PARIS2 reveals dynamic and long-range structures in the 5′UTR of two EV-D68 strains. a** PARIS2 validates predicted structures (black arcs) and reveals new alternative conformations (colored arcs) in the 699 nt 5′UTR of the US47 strain of EV-D68. 5′CL cloverleaf structure, IRES internal ribosome entry site, including domains II–VI, a1–a5 five alternative conformations. Numbers in parentheses are the counts of gapped alignments in each DG. **b** Long-range structures connecting the 5′UTR to the rest of the genome in two EV-D68 strains. All DGs have cov ≥ 0.01 for these two samples. The two stars indicate the right-side anchors for the strongest long-range strucgtures in the two strains, near the CRE and the region near the stop codon at the end of peptide 3D (3Dstop). **c** Long-range structures in the 5′UTR are primarily anchored in three linker regions L1–L3 (e.g., L1 is after domain I). **d** Diagram showing the three strongest long-range structures L1-CRE (connecting L1 and near CRE, in US47), L2L3-start (start codon, in US47), and L1L2-3Dstop (in VR1197). **e–g** DGs supporting the three long-range structures. In L1-CRE (**e**), the first ~5000 nt of the genome is shown. In the L2L3-start (**f**), the first ~800 nt of the genome is shown. In L1L2-3Dstop (**g**), the entire genome is shown. **h** Quantifying the strength of DGs for the three structures in all conditions. H HeLa cells, S SH-SY5Y cells, U US47, V VR1197, A AG, G GA. **i** Stability of the three structures, measured in minimal free energy (MFE), in the two strains.

in two strains are consistent with the abundance of PARIS2 measured duplexes, suggesting that genetics drive the formation of alternative conformations during EV-D68 evolution (Fig. 6i and Supplementary Fig. 33j–m). Together, the combined PARIS2 analysis and phylogenetic analysis revealed a dynamic model of the IRES structure, in the context of the entire genome and in evolution, setting the stage for further functional studies.

## Discussion

PARIS2 is highly efficient and sensitive, overcoming fundamental bottlenecks in current crosslink-ligation methods for RNA structure/interaction analysis. In addition, the mechanistic insights, and photochemical and enzymatic approaches are generally applicable in molecular biology. We discover surprisingly high hydrophobicity in crosslinked RNA that renders the classical

AGPC method inefficient; and develop a method capable of complete RNA recovery. The full recovery of intact crosslinked RNA enables targeted analysis of structures and interactions, as demonstrated in three applications on cellular and viral RNAs. The general property of crosslink-induced RNA hydrophobicity is reminiscent of crosslinked RNA–protein complexes that partition to the interphase[39–42]; however, it represents a distinct new mechanism. Here the RNA structure itself, coupled with the low pH, is the determinant of hydrophobicity, in contrast to RNA–protein crosslinks where the nonpolar amino acid residues control the hydrophobic behavior. The abnormal in vitro phase partition may be relevant to the role of RNA in phase separation in vivo, even though they occur at different physicochemical environments[43,44]. The newly developed DD2D gel method is robust and isolates crosslinked RNA fragments with high purity, outperforming alternative approaches, such as RNase R digestion and biotin-tagging of psoralens. To our knowledge, the TNA and DD2D gels are the only methods to completely and specifically recover total and crosslinked RNA. Our in-depth analysis of the photochemical damages in RNA during both the crosslinking and reversal steps have led to understanding of these processes. Based on these mechanistic studies, we introduce the first chemical and enzymatic approaches that enable the efficient prevention and bypass of these damages, solving long-standing problems in the photochemistry and RNA fields[14]. The prevention and bypass of photochemical damage will be useful in many RNA studies, since UV irradiation is a commonly used technique, such as in protein–RNA interaction analysis[45].

Despite these major improvements, there are still several steps that will benefit from further optimizations. For example, faster reacting crosslinkers will enable the analysis of more dynamic structures in vivo. The T4 RNA ligase proximity ligation only produces ~10% gapped reads; more efficient ligation will greatly increase the percentage of useful reads and the sensitivity of the method. New crosslinkers that overcome the psoralen (both AMT and amotosalen) bias toward uridines will enable the analysis of previously uncrosslinkable duplexes and further improve crosslink-ligation-based methods.

Using PARIS2 and an improved analysis pipeline, we found that the psoralen crosslinking captures ribosome SSU binding sites across the transcriptome. The bias toward uridines in psoralen crosslinking may confound analysis of translation, however, it is unlikely to be critical given the near uniform distribution of the uridines in most mRNAs. Alternatively, a uridine-abundance-based correction can be applied to obtain unbiased measurement of SSU binding sites. The simultaneous measurement of mRNA secondary structure and translation status in one experiment will make it possible to directly analyze the impact of RNA structures on translation, which has been a challenge due to mRNA structure dynamics and heterogeneity[46].

The improved PARIS2 and computational tools revealed a dynamic network of RNA structures and interactions involved in ribosome biogenesis and neurological disorder LCC, highlighting the power of integrating PARIS2 with human genetics in studying the consequences of noncoding mutations. Many noncoding mutations in genetic disorders and cancers remain difficult to study[47]. Direct determination of RNA structures and interactions models made it possible to raise hypothesis about the molecular effects of disease mutations and connections to specific cellular processes.

Based on the improved PARIS2, we built the first in vivo structure model for the EV-D68 RNA genome, revealing a complex global architecture and dynamic conformations. In particular, we found that antiviral drug treatment (e.g., blocking replication) reduces global long-range structures and specific structures in the 5′UTR (the L2L3-start duplex), suggesting that some of these long-range structures may contribute to virus

fitness. The conservation of a subset of the PARIS2-derived duplexes further support the functional relevance of newly discovered structures. The complete recovery and high coverage of viral RNAs by the improved photochemistry are essential for detecting the full spectrum of dynamic conformations in cells. Our identification of highly dynamic structures challenges the conventional view that RNA folds into global lowest energy states. Instead, conversions among suboptimal alternative conformations are likely to be frequent, and underlying transitions among functional states and infection stages. Further studies of each stage of the life cycle may reveal new functional structures that play important roles in the transitions among stages.

We report the first structure models of both the (+) and (−) strands for RNA viruses, revealing critical differences between the two strands associated with the asymmetric replication process. Given the critical role of RNA structures in template-switching induced genome recombination, simultaneous measurement of structures on both strands will lead to new insights into the mechanisms driving the generation of recombination hotspots, virus evolution, and emergence of new pathogenic strains[48].

Despite over three decades of research on IRES, our understanding of their in vivo dynamics remains limited. Others have found that certain host factors can induce minor conformations changes in IRES structures[49]. Our discovery of the extensive alternative conformations in the 5′UTR highlights the power of PARIS2. The dynamic conformations in domains V and VI, which bind the major translation initiation factors, including eIF3, 4B, 4G, and PTBP[50] are highly conserved in evolution, suggesting yet unknown functions of these conformations. We propose that the alternative conformations may represent different stages in the life cycle, such as translation, replication and packaging, or different stages in translation initiation.

Together, PARIS2 will enable RNA structurome and interactome analysis in increasingly more challenging biological systems, and enable functional and mechanistic investigations of RNA-centric regulations in normal physiology and human diseases. Understanding the structural basis of RNA functions in vivo will be critical for future drug development targeting RNAs.

## Methods

**Synthesis and characterization of amotosalen HCl.** Psoralen is the only class of reversible nucleic acid crosslinkers that can be used in mild physiological conditions, and AMT is the most commonly used one due to its relatively high solubility at 1 mg/ml in aqueous solutions (~3 mM). Nevertheless, crosslinking at 0.5 mg/ml does not approach saturation and therefore the solubility still limits its efficiency[11]. It is likely that this limited solubility is responsible for the low crosslinking efficiency (0.2–0.5% crosslinked RNA from total RNA)[9]. In a related class of methods that analyzes nucleotide flexibility/accessibility, as exemplified by SHAPE and DMS-seq, the RNA-reactive compounds are typically used at much higher concentrations to merely obtain single hit kinetics (e.g., 100 mM or higher for NAI-N3, and 650 mM for DMS[51,52]. In the chemical probing experiments, the reactions would destabilize RNA structures and therefore modifications should be limited to <1 per ~100 nt. However, in the case of crosslinking, RNA structures are stabilized, and therefore higher crosslinking efficiency does not have adverse effects.

One way to improve PARIS is to use psoralen derivatives that are more water soluble. Previous studies have shown that amotosalen (also known as S59 or S-59) is soluble at 50 mg/ml in aqueous solutions[12,53]. Amotosalen (compound 2 in patent US5,654,443) was used at 50 μg/ml, irradiated with 3 J/cm² 365 nm UV for inactivation of viruses and bacteria. The activity of amotosalen was slightly better than AMT at the same concentration[53]. The synthesis of amotosalen was described on page 44 of patent US5,654,443, but the procedure is unnecessarily complex. We synthesized amotosalen from trioxalen using a simplified three-step procedure as follows (see Supplementary Fig. 1).

Trioxalen + $ClCH_2OCH_3$ → CMT + methanol;
CMT + Boc-ethanolamine → Boc-amotosalen → amotosalen + Boc.

*General.* All chemicals for synthesis were obtained from commercial sources and used as received unless stated otherwise. Solvents were reagent grade. Thin-layer chromatography was performed using commercial Kieselgel 60, F254 silica gel plates. Flash chromatography was performed on silica gel (40–63 μm, 230–400 mesh). Drying of solutions was performed with $MgSO_4$ and solvents were removed

with a rotary evaporator. Chemical shifts for NMR measurements were determined relative to the residual solvent peaks ($\delta_H$ 7.26 for CHCl$_3$ and 2.50 for DMSO, $\delta_C$ 77.0 for CHCl$_3$, and 40.0 for DMSO). The following abbreviations are used to indicate signal multiplicity: s, singlet; d, doublet; t, triplet; q, quartet; m, multiplet; brs, broad signal; appt, apparent triplet.

## 3-(Chloromethyl)-2,5,9-trimethyl-7H-furo[3,2-g]chromen-7-one (2, CMT, or chloromethyl trioxalen, or 4′-chloromethyl-4,5′,8-trimethyl psoralen)

Compound **2** was synthesized as previously reported[54]. Trioxsalen (1.9 g, 4.4 mmol) was dissolved in AcOH by gently heating after which the solution was cooled back to room temperature. Chloromethyl methylether (16.0 g, 200 mmol) was added and the resulting reaction mixture was stirred at room temperature for 24 h. Next, more chloromethyl methylether (16.0 g, 200 mmol) was added and the solution was stirred at 35 °C. for 48 h. The reaction was cooled down to room temperature and allowed to stand for another 24 h. The formed precipitate was filtered off yielding 1.5 g (65%) of a white cotton-like solid. $^1$H NMR (400 MHz, CDCl$_3$) $\delta$ 7.60 (s, 1H), 6.27 (s, 1H), 4.74 (s, 2H), 2.58 (s, 3H), 2.54–2.52 (m, 6H).

## 2,2,2-Trifluoro-N-(2-((2,5,9-trimethyl-7-oxo-7H-furo[3,2-g]chromen-3-yl)methoxy)ethyl)acetamide (3, Boc-amotosalen)

The conversion of CMT to amotosalen can be accomplished with a Williamson ether synthesis method. Compound **2** (1.5 g, 5.4 mmol) was mixed with N-(2-hydroxyethyl)trifluoroacetamide (3.0 g, 19.1 mmol) and heated for 1 h at 100 °C. The mixture was cooled down to room temperature and recrystallized from methanol yielding an off-white powder. $^1$H NMR (400 MHz, DMSO) $\delta$ 8.30 (s, 1H), 7.70 (s, 1H), 6.31 (s, 1H), 4.62 (s, 2H), 3.52 (t, $J = 5.6$ Hz, 2H), 3.35 (t, $J = 5.4$ Hz, 2H), 2.46 (s, 6H), 2.43 (s, 3H).

## 3-((2-Aminoethoxy)methyl)-2,5,9-trimethyl-7H-furo[3,2-g]chromen-7-one hydrochloride (amotosalen HCl) (1)

Compound **3** was dissolved in 0.5 M Cs$_2$CO$_3$ in methanol and stirred at room temperature for 16 h. The mixture was concentrated in vacuo and purified using flash chromatography (DCM:MeOH, 9:1) yielding yellow crystals. The product was dissolved in ethanol and the mixture was cooled on an ice bath. A total of 1 M HCl in diethyl ether was added and the mixture was stirred for 4 h on ice. The white precipitate was collected by filtration yielding Amotosalen HCl. $^1$H NMR (400 MHz, DMSO) $\delta$ 8.02 (s, 3H), 7.80 (s, 1H), 6.34 (s, 1H), 4.68 (s, 2H), 3.62 (t, $J = 5.1$ Hz, 2H), 2.97 (d, $J = 4.9$ Hz, 2H), 2.52–2.44 (m, 9H).

**Measuring amotosalen solubility.** Solubility of the newly synthesized amotosalen was tested in water, PBS, and various other solutions. Amotosalen was previously reported to be soluble at least 50 mg/ml in 0.9% NaCl (ref. [12]). We dissolved amotosalen HCl in water so that there was a large amount of insoluble solid and the solution was saturated. The saturated solution has a bright orange color. We diluted the solution 2500-fold and observed an absorbance of 7.29 at 250 nm. This corresponds to 229 mg/ml at room temperature, given the specific absorbance of 26,900 M/cm (similar to AMT, 25,000 M/cm, and 8-MOP, 22,900 M/cm)[55]. We found that amotosalen HCl is soluble in 1× PBS pH 7.4 >100 mg/ml (did not push it to the limit). However, amotosalen HCl is partially insoluble at 10 mg/ml in the following solutions: 150 mM NaCl without buffer, 100 mM CH$_3$COONa pH 5.2, and highly insoluble in 1% SDS. These tests suggest that amotosalen is incompatible with ionic solutions, except the 100 mg/ml solution in PBS.

**Cells and animals.** HEK293 (ATCC, CRL-3216), HeLa (ATCC, CCL-2), and SH-SY5Y (ATCC, CRL-2266) cells were purchased from ATCC. HEK293 and HeLa were maintained in Dulbecco's modified Eagle's medium (DMEM, Gibco, 11965118) + 10% fetal bovine serum (FBS, Gibco, 10082147) + penicillin–streptomycin (Gibco, 15140163), in 37 °C incubator with 5% CO$_2$. SH-SY5Y was maintained in 1:1 mixture of Eagle's Minimum Essential Medium and F12 Medium (ATCC, 30-2003) + 10% FBS + penicillin–streptomycin. Wild-type C57BL/6J mice were bred and maintained under specific pathogen-free conditions, fed standard laboratory chow, and kept on a 12-h light/dark cycle and temperature and humidity were kept at 22 ± 1 °C, 55 ± 5%. C57BL/6J female or male mice aged 4–6 weeks old were used for all experiments. All cell culture were handled according to protocols approved by the University of Southern California. All animals were used according to animal use protocols granted by the Institutional Animal Care and Use Committee at the University of Southern California.

## Crosslinking

*Crosslinking of cells.* AMT (Sigma-Aldrich A4330) and amotosalen were dissolved in pure water at a concentration of 1 and 100 mg/ml, respectively. Cells cultured to 80% confluency in 10 cm dish were washed twice with 1× PBS, and then were treated with 0.5 mg/ml AMT, 0.5, 2.0, or 5.0 mg/ml amotosalen in 1× PBS for 15 min in 37 °C incubator. Control cells were incubated in 1× PBS. The cells in crosslinking solution were placed on ice trays in Stratalink 2400 UV crosslinker and crosslinked for 30 min under UV$_{365 nm}$ bulbs[13]. Swirl the plates every 10 min

and make sure that plated are horizontal. Remove crosslinking solution after crosslinking and wash cells twice with 1× PBS.

*Crosslinking of tissues.* Four mice brain tissues were harvested and placed in ice-cold HBSS (Gibco, 14025076). The tissues were dissociated by passing through 5 ml pipet 20 times. After three times washing with 1× ice-cold HBSS, tissues were resuspended in 2 ml 0.5 mg/ml amotosalen and incubated for 15 min in dark. Tissues in crosslinking solution were placed on ice trays and crosslinked for 30 min under UV365nm bulbs.

*Crosslinking of nucleic acid strands.* DNA oligos, RNA oligos, or total RNA samples were incubated with specific concentration of AMT or amotosalen in 1× PBS for 5 min. Oligo or RNA samples in crosslinking solution were transferred to a clean surface with ice beneath it and placed in Stratalink 2400 UV crosslinker. Samples were crosslinked for 30 min under UV$_{365 nm}$ bulbs.

## Extraction of crosslinked RNA

*TNA method.* For each 10 cm dish cells, added 100 µl of 6 M GuSCN (Sigma, 368975) and lysed cells with vigorous manual shaking for 1 min. After cell were lysed into a nearly homogenous solution, cell lysate was added 12 µl of 500 mM EDTA EDTA (Invitrogen, 15575020), 60 µl of 10× PBS (Invitrogen, AM9625), and water to final volume of 600 µl. Then each sample was passed through a 25 G or 26 G needle ~20 times to further break the insoluble material. PK (Thermo Scientific, EO0492) was added to final concentration of 1 mg/ml, and PK treatment was performed at 37 °C for 1 h on a shaker. After PK digestion, 60 µl of 3 M sodium acetate (pH 5.3; Invitrogen, AM9740), 600 µl of water-saturated phenol (pH 6.7; Invitrogen, AM9712), and one volume pure isopropanol were added to precipitate total nucleic acids by spinning at 12,000 × g for 20 min at 4 °C. After twice washing using 70% ethanol, total nucleic acids were resuspended in 300 µl of nuclease-free water (Supplementary Method).

For 100 µg of TNA samples, 50 units of TURBO™ DNase (Invitrogen, AM2239) were added to remove DNA at 37 °C for 20 min. Then added 20 µl of 3 M sodium acetate (pH 5.3), equal volume of water-saturated phenol (pH 6.7), two volume of pure isopropanol to precipitate RNA sample by spinning 20 min at 12,000 × g at 4 °C. To compare the recovery efficiency, crosslinked RNAs were also extracted using TRIzol reagent and RNeasy Mini™ kit (Qiagen, 74104), according to the manufacturer's instructions.

The PK digestion should clarify the solutions to some extent and greatly reduce turbidity. The addition of isopropanol should clarify the solution, resulting in obvious compact and stringy precipitates that contain both DNA and RNA, but little protein. Most of the TNA sample should be soluble. If there is still some insoluble material, spin down and remove it. The A$_{260}$/A$_{280}$ ratios of crosslinked TNA samples are usually in ~1.90, in the middle between the ratios for DNA and RNA. The A$_{260}$/A$_{230}$ ratios for the control samples are usually >2.1 and the ratios for crosslinked samples are usually <1.9. The TapeStation profile for the TNA from crosslinked samples should show an obvious smear across the entire size range, while controls show three major peaks, namely the small RNAs, the 18S and 28S rRNAs. The controls should have a RIN number close to 10, while the crosslinked ones have a RIN number <8.

**RNA fragmentation.** Crosslinked RNAs were fragmented using ShortCut RNase III (NEB, M0245). Briefly, 10 µg of crosslinked RNA was fragmented using 10 µl of RNase III with 50 mM MnCl$_2$ and 1× supplied shortcut buffer at 37 °C for 5 min. After fragmentation, equal volume of phenol was immediately added to stop the reaction. Then one-tenth volume of 3 M sodium acetate (pH 5.3), 3 µl of GlycoBlue (Invitrogen, AM9516), three volume of pure ethanol were added to precipitate RNA. Fragmented RNA was resuspended in RNase-free water and checked size distribution using TapeStation. Different fragmentation conditions also were tested in this study, such as different RNase III amount, different fragmentation time, and different concentration of MnCl$_2$.

After 5 min of shortcut digestion, reaction need to be stopped as soon as possible to get the optimal size distribution. Longer reaction time will reduce the RNA fragments size. The size distribution of fragmented crosslinked RNA can be analyzed by Bioanalyzer or TapeStation system (Agilent TapeStation Software v3.2). If using TapeStation, high-sensitivity D1000 ScreenTape plus high-sensitivity RNA sample buffer should be used because of short RNA size after fragmentation.

**DD2D purification of crosslinked RNA (dsRNA fragments)**
*First-dimension gel.* Prepare 8% 1.5 mm thick denatured first-dimension gel using the UreaGel system (National Diagnostics, EC-833). Loading dsRNA ladder (NEB, N0363S) as molecular weight marker. Run the first-dimension gel at 30 W for 7–8 min in 0.5× TBE (Invitrogen, 15581044). After electrophoresis was finished, staining the gel with SYBR Gold (Invitrogen, S11494) in 0.5× TBE and excising each lane between 50 nt to topside from the first-dimension gel. The second-dimension gel can usually accommodate three gel splices.

*Second-dimension gel.* Prepare the 16% 1.5 mm thick urea denatured second-dimension gel using the UreaGel system[56]. Using prewarmed 0.5× TBE buffer to fill the electrophoresis chamber to facilitate denaturation of the crosslinked RNA.

Run the second dimension at 30 W for 50 min to maintain high temperature and promote denaturation. DD2D gels were acquired and analyzed by Bio-Rad Image System (Image Lab software, v6.0.1) or iBright FL1500 Image System (iBright Analysis Software, v3.1.2). Gel containing the crosslinked RNA above the diagonal from the 2D gel was excised and crushed for RNA extraction (Supplementary Note 3). The different combination of 6, 8, and 10 first-dimension gel, and 16 and 22.5% secondary dimension gel was also tested in this study.

Fifteen-well combs should be used for the first-dimension gel so that each lane is narrower and the second dimension has a higher resolution. No >10 μg of fragmented RNA should be loaded to each line. A 300 nm transillumination should be used to image the gel (254 nm epi-illumination will reverse the psoralen crosslinking. To make the second-dimension gel, put the square plate horizontally and arrange gel slices in a "head-to-toe" manner with 2–5 mm gap between them. Apply 20–50 μl 0.5× TBE buffer on each gel slice to avoid air bubbles when placing the notched plate on top of the gel slices. Remove the excess TBE buffer after the cassette is assembled. Pour the gel solution from the bottom of the plates, while slightly tilting the plates to one side to avoid air bubbles building up between the plates. If there are air bubbles, use the thin loading tips to draw them out. During the second-dimension gel running, the voltage started ~300 V and gradually increased to 500 V, while the current started ~100 mA and gradually decreased to 60 mA.

**Proximity ligation.** Purified dsRNA fragments were proximity ligated by T4 RNA ligase 1 (NEB, M0437M). Briefly, 2 μl of 10× ligation buffer, 5 μl of T4 RNA ligase, 1 μl of SuperaseIn (Invitrogen, AM2696), and 1 μl of 0.1 mM ATP were added to 10 μl of purified dsRNA fragments. Ligation mixture was incubated at room temperature overnight. After ligation, the samples were boiled for 2 min to stop the reaction. After heat denaturation, samples were centrifuged to remove the precipitate and then precipitated by ethanol.

**Reverse crosslinking.** Proximity ligated RNA fragments were placed on a clean surface with ice beneath it. To protect RNA from UVC damage, 2 μl of 2.5 mM AO (Fisher Scientific, AC300911000) was added to each sample (total volume 20 μl). Samples were irradiated with $UV_{254\,nm}$ for 30 min. After reverse crosslinking, RNA was purified with three volume of ethanol and 1 μl of GlycoBlue (Invitrogen, AM9516).

**Adapter ligation.** Reverse-crosslinked RNAs were heated at 80 °C for 90 s, then snapped cooling on ice. To each sample, 3 μl of 10 μM ddc adapter (/5rApp/AGATCGGAAGAGCGGTTCAG/3ddC/; IDT; Supplementary Table 5), 1 μl of T4 RNA ligase 1, 2 μl of DMSO, 5 μl of PEG8000, 1 μl of 0.1 M DTT, 1 μl of SuperaseIn, and 2 μl of 10× T4 RNA ligase buffer were added to perform adapter libation at room temperature for 3 h. After adapter ligation, following reagents were added to remove free adapters: 3 μl of 10× RecJf buffer (NEBuffer 2, B7002S), 2 μl of RecJf (NEB, M0264S), 1 μl of 5'deadenylase (NEB, M0331S), 1 μl of SuperaseIn, Reaction was incubated at 37 °C for 1 h. Then, 20 μl of water was added to each sample to make total volume of 50 μl and Zymo RNA clean and Concentrator-5 (Zymo Reasearch, R1013) was used to purify RNA.

**Reverse transcription.** SSIV (Invitrogen™, 18090010) was used to performing reverse transcription. The reaction buffer was optimized $Mn^{2+}$ buffer (1×): 50 mM Tris-HCl (PH 8.3), 75 mM $CH_3COOK$, and 1.5 mM $MnCl_2$. A total of 1 pmol of barcoded RT primer (Supplementary Table 5) and 1 μl of 10 mM dNTP were added to RNA samples and heated at 65 °C for 5 min in a PCR block, chill the samples one ice rapidly. Then, 4 μl of 5× $Mn^{2+}$ buffer, 2 μl of 0.1 M DTT, 1 μl of SuperaseIn, and 1 μl of SSIV were added to each sample. Mixed sample was incubated at 25 °C for 15 min, 42 °C for 10 h, 80 °C for 10 min; hold at 10 °C. After reverse transcription, 1 μl RNase H and RNase A/T1 mix were added and incubated at 37 °C for 30 min in a thermomixer to remove RNA. Synthesized cDNAs were purified using Zymo DNA clean and Concentrator-5 (Zymo Reasearch, R41013).

**cDNA circularization and library generation.** A total of 1 μl of CircLigase™ II ssDNA Ligase (Lucigen, CL9021K), 1 μl of 50 mM $MnCl_2$, and 10× CircLigaseII™ buffer were added to cDNA sample and performed circularization at 60 °C for 100 min. A 80 °C treatment for 10 min was followed to stop the reaction. The circularized cDNA products were directly used to library PCR. Library PCR preparation was done, as described in ref. [57]. PCR products were run on 6% native TBE gel. Gel containing DNA products from 175 bp and topside (corresponding to >40 bp insert) was excised and crushed for DNA extraction.

**UVC damage prevention.** A total of 200 ng of RNA and cDNA were irradiated with $UV_{254\,nm}$ for 10 and 30 min to introduce the UVC damage. cDNA sample is generated from total RNA of HEK293T cells. UVC damages were determined by ct value of RT-qPCR. Different concentration of AO (Fisher Scientific, AC300911000), EB (Invitrogen, 15585011), proflavine (Sigma, P2508-1G), acetone (Sigma, 650501-1 L), and SYBR Gold (Invitrogen, S11494) were added to each sample to test their UVC prevention efficiency. Other conditions were also tested in this study, such as high salt concentration (1 M NaCl), denaturing agents 4 M Urea and 50% formamide (Thermo Scientific, 17899).

**Effects of antioxidants on PUVA damages.** Following antioxidants were used to test RNA protection from PUVA damages. $O_2^{\bullet-}$ scavenger: tiron (Sigma, 172553) and MnTBAP (Sigma, 475870); •OH scavenger: mannitol (Sigma, M4125), DMSO (Sigma, D2650), and glycerol (Sigma, G5516); $^1O_2$ scavenger: $NaN_3$ (Sigma, S2002); and general radical scavenger: vitamin C (Sigma, 11140). Cells cultured to 70% confluency were treated with normal culture media with 0.5 mg/ml AMT and different concentration of antioxidants for 15 min in dark. After incubation, the media was replaced with 0.5 mg/ml AMT plus different antioxidants. Control cells were incubated with 1× PBS. The plates in crosslinking solution were placed on ice bed in UV crosslinker for 30 min crosslinking. After crosslinking, total RNA was extracted by TNA method. Crosslinking efficiency was analyzed by DD2D gel system. PUVA damage was determined by ct value of RT-qPCR.

**Primer extension.** A 48-mer RNA oligo (5'-CUUGCUAGGCCCGGGUUC-CUCCCGGGCCUAGCCCUGUCUGAGCGUCGC-3'; IDT; Supplementary Table 5) was crosslinked by 0.5 mg/ml AMT with $UV_{365\,nm}$ for 30 min, and was reverse crosslinked by $UV_{254\,nm}$ for 30 min with the protection of AO. Synthesis was primed by DNA primer (5'-GCGACGCTCAGACAGG-3'; IDT) annealed to the 3' end of RNA template. Unless otherwise specified, reverse transcription reactions were performed in 20 μl volumes. Before all primer extension assay, samples were treated by heating in 10 μl of solution containing 1 pmol of RNA template, 1 pmol of DNA primer, and 0.5 mM dNTP (no dNTP for TGIRT™-III Enzyme) at 65 °C for 5 min, then snap cooling on ice at least for 1 min. Following enzymes were used to extension. SSII (Invitrogen, 18064022), SSIII (Invitrogen, 18080093), SSIV, TGIRT™-III Enzyme (TGIRT; Ingex, TGIRT50), HIV recombinant reverse transcriptase (HIV; Worthington, LS05006).

*SSII*: to each was added 4 μl of 5× standard reaction buffer (250 mM Tris-HCl pH 8.3, 375 mM KCl, and 15 mM $MgCl_2$) or 5× $Mn^{2+}$ buffer (250 mM Tris-HCl pH 8.3, 375 mM KCl, and 15 mM $MnCl_2$), 2 μl of 0.1 M DTT, 1 μl of SuperaseIn, and 50 units of SSII. Samples were mixed and incubated at 42 °C for 60 min, followed by 80 °C for 10 min to stop the reaction.

*SSIII*: hybridized primer–template was added 4 μl of 5× standard reaction buffer (250 mM Tris-HCl pH 8.3, 375 mM KCl, and 15 mM $MgCl_2$) or 5× $Mn^{2+}$ buffer (250 mM Tris-HCl pH 8.3, 375 mM KCl, and 15 mM $MnCl_2$), 2 μl of 0.1 M DTT, 1 μl of SuperaseIn, and 50 units of SSIII. Then mixed samples were incubated at 42 °C for 5 min, 50 °C for 60 min, followed by 80 °C for 10 min to stop the reaction.

*SSIV*: to each was added 4 μl of 5× commercial buffer or 5× $Mn^{2+}$ buffer (250 mM Tris-HCl pH 8.3, 375 mM KCl, and 2.5/7.5/15 mM $MnCl_2$), 2 μl of 0.1 M DTT, 1 μl of SuperaseIn, and 50 units of SSIV. Samples were incubated at 42 °C 5 min, 55 °C for 60 min, followed by 80 °C for 10 min to stop the reaction.

*TGIRT*: primer–template sample was added 4 μl of 5× standard reaction buffer (100 mM Tris-HCl pH 7.5, 250 mM NaCl, and 50 mM $MgCl_2$) or 5× $Mn^{2+}$ buffer (100 mM Tris-HCl pH 7.5, 250 mM NaCl, and 15 mM $MnCl_2$), 2 μl of 0.1 M DTT, 1 μl of SuperaseIn, and 50 units of TGIRT. Sample were mixed and incubated at 42 °C for 30 min. Then 2.5 μl of 10 mM dNTPs were added to reaction and incubated at 60 °C for 2 h (ref. [58]).

*HIV*: to each was added 4 μl of 5× standard buffer (250 mM Tris-HCl pH 8.3, 375 mM $CH_3COOK$, and 250 mM $MgCl_2$) or 5× $Mn^{2+}$ buffer (250 mM Tris-HCl pH 8.3, 375 mM $CH_3COOK$, and 15 mM $MnCl_2$), 2 μl of 0.1 M DTT, 1 μl of SuperaseIn, and 50 units of HIV. Samples were incubated at 42 °C 5 min, 55 °C for 60 min, followed by 80 °C for 10 min to stop the reaction.

After extension, 1 μl of 5 M NaOH was added to each tube and incubate the tubes for 3 min at 95 °C. Then samples were neutralized with 1 μl of 5 M HCl. After purification with ethanol precipitation, cDNA products were separated on 20% denaturing polyacrylamide gels. The Gene Ruler was 10-bp DNA ladder (Invitrogen™, 10821015).

**ROC analysis.** The cryo-EM model of the 28S rRNA was downloaded from RCSB Protein Data Bank (PDB; ID: 4V6X). Watson–Crick and non-Watson–Crick base pairs were analyzed using the DSSR software (v1.7.7)[59]. The structure information between every 5-nt bin in 28S rRNA was identified and was used as a gold standard to evaluate the performance of PARIS2 in detecting 2D structures. The true-positive datasets were defined by 5-nt pairwise with >3 nt were base paired, and the true-negative datasets were otherwise. For PARIS2 sequence data, base pair information of each pairwise 5-nt were also calculated. We classified the PARIS2 pairwise as true-positive interaction or false-positive interaction on the basis of whether the PARIS2 pairwise was same with cryo-EM model of 28S rRNA. The percentage of normalized reads number of each pairwise were used as the score of each PARIS2 pairwise. The ROC curve was obtained by varying the threshold of PARIS2 pairwise score from 0 to 1 and counting the true-positive rate and false-positive rate.

**Melting curve analysis.** For each oligonucleotide annealing reaction, 10 pmol of each wild type (WT) and 3G > A mutation (Mut) oligonucleotide were combined with hybridization buffer (5 mM NaCl, 10 mM Tris-HCl pH 7.5, and 1 mM EDTA), 1 μl of 20× SYBR Green (Invitrogen™, S7563) and made up to a final

volume of 20 μl with H₂O. Samples were loaded to a 96-well plate in triplicate. Samples were heated at 95 °C for 3 min and then rapidly cooled to 20 °C held for 10 min to facilitate annealing of the oligonucleotides. A single fluorometric data point was collected for every 1 °C increment as the temperature was raised back to 90 °C. Raw high fluorometric data were exported from the ABI 7300 into Microsoft Excel. The mean fluorescent value was calculated from the combined triplicates samples. The negative first derivative of each fluorescence acquisition point was generated using the following Excel equation[60]:

$$-dF/dT = \text{Sum}(((\text{Log}(Xn)) - (\text{Log}(Xn - 1)))^* - 1)$$

where $Xn$ is equal to a particular $y$-axis fluorescence value, and $Xn - 1$ is equal to the fluorescence value preceding the $Xn$ $y$-axis value. The mean of all negative first derivatives for each annealing temperature was plotted to determine the $T_m$.

**Virus stock preparation**. EV-D68 (US/MO/14-18947 strain, US/MO/47 for short in the paper, ATCC, VR-1823) stocks were prepared by infecting HeLa cells at 33 °C in 5% CO₂ for 3–4 days until obvious CPE (cell rounding and sloughing) was observed. Then infected cells were subjected to three freezing–thawing cycles followed by centrifugation to remove cell debris. Virus titers were determined by 50% tissue culture infective dose (TCID50) assay and calculated by the Reed and Muench method. The virus stocks were stored at −80 °C for use.

**Design of biotinylated antisense oligos**. Seven antisense oligos per genome RNA were designed (ChIRP Probe Designer, https://www.biosearchtech.com/support/tools/design-software/chirp-probe-designer) for targeting EV-D68 US/MO/47 or VR1197 strains, 20 nt for each oligo (Supplementary Table 5). The biotinylated oligos were ordered from IDT reconstituted at 100 mM in water. An equimolar mixture of all the antisense probes were made without further dilution, therefore it is 100 μM total concentration. The dissolved biotinylated antisense oligos were stored at −80 °C for use.

**One-step growth curve of EV-D68**. HeLa and SH-SY5Y cells were cultured in 12-well plates until cell confluence reached 80–90%, followed by inoculation of EV-D68 (US/MO/47 or VR1197) at an MOI of 5 and adsorption of virus at 4 °C for 1 h. Then, the supernatant was removed and cells were washed twice with 1× PBS, followed by addition of 1 ml DMEM (HeLa)/Opti-MEM (SH-SY5Y) + 2% FBS per well before being incubated at 33 °C. At 0, 1, 3, 6, 9, 12, 18, and 24 h.p.i., the supernatant and cells from each well were collected separately. The collected cells were subjected to three freezing–thawing cycles followed by centrifugation to remove cell debris. The virus titer was determined by TCID50 assay.

**Immunostaining**. HeLa cells were grown to 50–70% confluence in a 24-well plate. For studies involving EV-D68 infection, cells were infected with EV-D68 at an MOI of 1 for 18 h at 33 °C. Samples were fixed with 4% paraformaldehyde (Sigma, 1004960700) and stored at room temperature for 10 min, and washed with PBS three times followed by 0.1% Triton-X (Sigma, X100) incubation at 4 °C overnight. Cells were washed three times in 1× PBS before blocking with 2% bovine serum albumin in PBS for 1 h. Then cells were incubated with rabbit polyclonal anti-VP1 of EV-D68 (GeneTex, GTX132313) at a final concentration of 1 μg/ml at room temperature for 1 h. Wash three times with PBS. Then a secondary goat anti-rabbit rhodamine red-X (Thermo Fisher, R-6394) at a final concentration of 1 μg/ml was added into cells for 1 h incubation. To visualize nuclei, DAPI stain (1:1000; Sigma, D9542) was added for 3–5 min incubation followed by three times washing. At last, images were taken with on a fluorescence microscope (Airyscan Confocal Microscope, ZEISS ZEN Imaging Software v3.2) using DAPI and rhodamine filters.

**Determination of inhibitor efficiency**. To test the efficiency of the selected inhibitors, rupintrivir (AG7088, 0.5 μM; AG for short in this paper) and geldanamycin (2 μM; GA for short in this paper), the inhibitors were added to cells at two different timepoints, 30 min preinfection and 12 h post infection, respectively. As a negative control, the same volume of DMSO was added. Briefly, HeLa cells were cultured in six-well plates until cell confluence reached 80–90%. Then, for one timepoint, the cells were treated with inhibitors for 30 min, followed by inoculation of EV-D68 at an MOI of 5 and incubation at 33 °C for 18 h. For the other timepoint, the cells were infected with EV-D68 for 12 h, followed by addition of inhibitors and incubation for another 6 h. The cells were collected for determination of viral RNA by RT-qPCR.

**Target RNA enrichment**
*mRNA enrichment*. mRNA was enriched from total crosslinked RNA using Poly(A) Purist™ MAG Kit (Invitrogen, AM1922), according to the manufacturer's instructions.

*Viral RNA enrichment*. Total RNA (200 μg) extracted from EV-D68 infected cells were mixed with the cocktail (2 μl, 100 μM) of seven biotinylated DNA oligos (IDT), which was maintained at 37 °C overnight with rotation in the hybridization buffer from ref. [61]. At the end of the hybridization, 100 μl of MyOne Streptavidin C1 Dynabeads (Invitrogen, 65002) were added into the RNA-probe hybridization

solution for additional 4 h rotation at 37 °C. After five times washing, beads were resuspended in 0.2 units/μl Turbo DNase at 37 °C for 20 min to degrade DNA probes followed by 80 °C 90 s treatment to release all target RNA as much as possible. Released RNA was separated from beads and purified with ethanol precipitation. To test the intactness and purity of enriched RNA, we performed gel electrophoresis on a 1% agarose gel, using ssRNA (NEB, N0362S) as the ladder. In addition, viral RNA profile was tested using TapeStation.

**Data analysis**
*Preparing the masked genome indices*. In order to accurately and easily analyze PARIS data, pseudogenes and multicopy genes from gencode, refGene, and Dfam were masked from hg38/mm10 genome. And then single copy of them was added back as a separated "chromosome". For example, multicopy of snRNAs were masked from the basic hg38/mm10 assembly genome, and nine snRNAs (U1, U2, U4, U5, U6, U11, U12, U4atac, and U6atac) were concatenated into one reference, separated by 100 nt "N"s, was added back. The curated hg38/mm10 genome contained 25 reference sequences, or "chromosomes", masked the multicopy genes and added back single copies. This reference is best suited for the PARIS analysis. The adjusted genome reference was used for mapping reads and IGV visualization. The EV_D68 viral genome (GenBank, KM851225.1) was downloaded from NCBI and manually corrected based on our viral sequencing data. After mutation identifying using GATK software (v4.1.9.0), three variant sites on EV_D68 genome were corrected (2023:G → A; 2647:G → A; 3242:A → G). The curated EV_D68 genome was added to hg38 refence as an independent chromosome.

*Mapping*. Sequencing data were preprocessed to remove adapters form the 3′ end using Trimmomatic (v0.36). PCR duplicates were removed using readCollapse script from the icSHAPE pipeline[57]. Then the library was split based on the barcodes using splitFastqLibrary from icSHAPE pipeline. 5′ Header was removed using Trimmomatic.

After primary preprocessing, reads were mapped to manually curated hg38 or mm10 genome using STAR program (v2.7.0 f)[62]. The parameters used are as follows: STAR --runThreadN 8 --runMode alignReads --genomeDir OuputPath --readFilesIn SampleFastq --outFileNamePrefix Outprefix --genomeLoad NoSharedMemory outReadsUnmapped Fastx --outFilterMultimapNmax 10 --outFilterScoreMinOverLread 0 --outSAMattributes All --outSAMtype BAM Unsorted SortedByCoordinate --alignIntronMin 1 --scoreGap 0 --scoreGapNoncan 0 --scoreGapGCAG 0 --scoreGapATAC 0 --scoreGenomicLengthLog2scale -1 --chimOutType WithinBAM HardClip --chimSegmentMin 5 --chimJunctionOverhangMin 5 --chimScoreJunctionNonGTAG 0 --chimScoreDropMax 80 --chimNonchimScoreDropMin 20.

*Classify alignments*. The primary mapping alignments were extracted from SampleAligned.sortedByCoord.out.bam (SAMtools, v1.8). Then the primary mapping alignments were filtered to remove low-confidence segments, rearranged and classified into six different types using gaptypes.py (https://github.com/zhipenglu/CRSSANT). cont.sam, continuous alignments; gap1.sam, noncontinuous alignments with one gap; gapm.sam, noncontinuous alignments with more than one gaps; trans.sam, noncontinuous alignments with the two arms on different strands or chromosomes; homo.sam, noncontinuous alignments with the two arms overlapping each other; bad.sam, noncontinuous alignments with complex combinations of indels and gaps. Gap1. and gapm alignments containing splicing junctions and short 1–2 nt gaps were filtered out using gapfilter.py (https://github.com/zhipenglu/CRSSANT) before further processing. Then filtered gap1.sam, filtered gapm.sam, and trans.sam were used to analyze RNA structures and interactions.

*Cluster alignments to groups*. Filtering alignments were assembled to DGs and NGs using the crssant.py script (https://github.com/zhipenglu/CRSSANT).

**Global profiling of ribosome small subunit analysis**. mRNA–rRNA interaction chimeric alignments were extracted using sam2mRNArRNAchimera.py (https://github.com/minjiezhang-usc). The mRNA–rRNA interaction alignment can be directly loaded to IGV to visualize the binding sites of mRNAs on the 45S unit. Upstream and downstream 200 nt windows of transcription start site and transcription termination site were extracted and used to analyze the binding sites of h18 and h26 on the meta mRNA (mRNAmegaCoverage.py script).

**Global profiling of spliceosomal snRNP binding sites**. snRNA–target interaction alignments were extracted using awk command (v4.2.0). snRNA–target interaction alignments with at least 15 nt matches for the snRNA targets were filtered using filterchimera.py. A total of 200 nt windows around splice sites was extracted for gencode gtf file using gtf2splice.py. Chimera connecting-specific snRNA regions were further extracted using sam2chimera.py script. Coverage along the 200 nt windows was calculated using bedtools coverage (v2.29.2). Meta-analysis for all windows around 5′ and 3′ splice sites were performed with windowmeta.py. The Output.bedgraph can be loaded to IGV (v2.8.13) for visualization.

**Analysis of EV-D68 RNA structure conservation**. A total of 508 complete genomic sequences of EV-D68 strains were retrieved from the NIAID Virus Pathogen Database and Analysis Resource (ViPR; http://www.viprbrc.org/). After removing duplicate sequences, 491 unique genomic sequences were remained. Manually curated US/MO/47 genome plus above 491 unique genomic sequences were used for alignment and further analysis. To analyze the structure conservation of EV-D68 RNA structures obtained from PARIS data, corresponding region of proposed structure from each sequence was extracted to perform multiple sequence alignment (MSA) using MSCULE (v3.8.31). The conservation of RNA secondary structure within each data set was evaluated, using RNAz (v2.0)[63] by calculating the $z$-score and the Structure Conservation Index. The following parameters were applied: --both-strands --no-shuffle. The scoring results and consensus structure were visualized by R-chie (https://www.e-rna.org/r-chie/). R-scape (v1.5.16)[64] was also used to study conserved RNA structure by measuring pairwise covariations observed in multiple sequence alignment (491 sequences plus curated US/MO/47). The following parameters were applied: --fold, the default $E$ value 0.05.

**Analysis of structure conservation in the 5′UTR in EVA/EVB/EVC**. For EVA genomes, 1484 sequences were aligned and 499 sequences (maximum sequence number for RNAz) were selected by applying a script called rnazSelectSeqs.pl involved in RNAz program, which were analyzed by running RNAz.

For EVB genomes, 379 sequences were aligned and analyzed using RNAz.

For EVC genomes, 741 sequences were aligned and 499 sequences (maximum sequence number for RNAz) were selected by applying a script called rnazSelectSeqs.pl involved in RNAz program, which were analyzed by running RNAz.

rnazSelectSeqs.pl is one of the scripts involved in RNAz program, which is used for optimization of mean pairwise identity for alignment (default: 80). In addition, too similar sequences (>99%) are removed.

**Design of RNA structure mutations using RNA2DMut**. RNA2DMut (https://rna2dmut.bb.iastate.edu)[65] can generate all possible point mutations of an input sequence and predict structural information based on the Boltzmann 2D structural ensemble. The sequences of all domains and alternative structures of EV-D68 IRES, were analyzed at each base to determine the effect of every possible point mutation on 2D structure. Based on RNA2DMut predictions, mutations were selected to disrupt each target RNA structure without affecting other IRES structures of EV-D68.

**Luciferase reporter assays**. To construct the mutants of IRES, base changes were introduced into wild-type IRES by mutagenic primers (Supplementary Table 5), followed by overlap PCR. The PCR products purified and digested with Kpn I (NEB, R0142S) and Not I (NEB, R0189S) were cloned into pcDNA3 RLUC POLIRES FLUC vector backbone (Addgene, 45642)[66]. As a negative control, a luciferase reporter vector with the original POLIRES replaced by a short linker sequence full of stop codons was included in our assay. All versions of IRESs were confirmed by Sanger sequencing. The sequences of primers used for cloning wild type and mutants of IRES, as well as the negative control, are shown in Supplementary Table 5.

A total of $1 \times 10^5$ HEK293T cells in 24-well culture plates were transiently transfected with 1 μg of each plasmid DNA using Lipofectamine 3000 (Invitrogen L3000015), according to the manufacturer's instructions. After 24 h, the cells were resuspended in 75 μl of DMEM and equal Dual-Glo Luciferase Assay Reagent was added to lyse cells for 10 min. Then, the firefly luminescence was measured. Next, 75 μl Dual-Glo Stop&Glo Reagent was added and after incubation for 10 min, Renilla luminescence was measured. The Renilla luciferase expression in transfected cells was used to correct for variations in transfection efficiency. The luciferase activities were measured on a BioTek Synergy H1 Hybrid Multi-Mode Reader (SoftMax® Pro GxP Software). All the luciferase assays were performed in triplicate.

**Reporting summary**. Further information on research design is available in the Nature Research Reporting Summary linked to this article.

## Data availability

The raw and processed PARIS2 sequencing data were deposited to Gene Expression Omnibus (GEO) with accession number GSE149493. The data supporting the findings of this study are available from the corresponding authors upon reasonable request. Source data are provided with this paper.

## Code availability

PARIS2 analysis scripts are available on GitHub partly at https://github.com/zhipenglu/CRSSANT and partly at https://github.com/minjiezhang-usc/PARIS2. The engineered genome reference of hg38/mm10 are available at https://drive.google.com/open?id=1wHSC-mf1jNNClXrVqMugqVmDVT4Crxzz.

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

## Acknowledgements

This work was supported by NIH. R00HG009662, and startup fund from USC to Z.L. Computation for the work described in this paper was supported by the University of Southern California's Center for High-Performance Computing (https://hpcc.usc.edu). We also acknowledge the USC Research Center for Liver Disease (P30DK48522) and Norris Comprehensive Cancer Center (P30CA014089) for their support of our research. We thank the USC Keck Genomics Platform and Illumina for high-throughput sequencing reagents and service. We thank A.G. Matera (UNC Chapel Hill) and H.Y. Chang (Stanford University) for the generous support of the early stages of technology development.

## Author contributions

Z.L. conceived this project and designed the overall PARIS2 strategy. M.Z., C.Y., and Z.L. developed the method. W.A.V. synthesized amotosalen. K.L. and J.B. performed the virus studies. M.L., R.V.D., W.H.L., M.L.C., and J.-F.C. participated in method optimizations and writing. M.Z., K.L., J.B., and Z.L. performed the analysis. Z.L. wrote the manuscript with input from all authors.

## Competing interests
The authors declare no competing interests.
