## [Peer Review File · Nature Communications]

REVIEWER COMMENTS

Reviewer #1 (Remarks to the Author):

The manuscript from Zhang M et al optimized several steps of the PARIS protocol to enable better capture of RNA-RNA interactions genome-wide. They then applied it to a variety of questions including translation, U8 interactions and the RNA interactome of enteroviruses. While a tremendous amount of work has been done, the manuscript comes across as scattered, with each sub-story being poorly developed. The authors may benefit from splitting this manuscript into better developed individual manuscripts.

Here are the major comments:

- 1) The authors developed amotosalen because they say that AMT is only soluble at 1 mg/ml. While efforts to develop a new psoralen compound is admirable, this reviewer is unsure why there is a need for this new compound in this protocol. AMT's solubility is at 1mg/ml. That is already highly soluble. Crosslinking experiments done at 0.5mg/ml inside cells are already so extensive that RNA is being crosslinked to proteins, resulting in a need to use proteinase K during RNA extraction. As such, more crosslinking is not necessarily better. There is a possibility of everything will be crosslinked to everything, making it difficult to determine which crosslinks are necessarily biologically relevant. Can the authors show the benefits of having even more crosslinking using amotosalen? Can they capture additional real crosslinks that were missing in PARIS? How many more real crosslinks are they capturing? What is the false positive rate?
- 2) Similar to point 1 above. The authors claim that the RNA duplexes captured by AMT and amotosalen are similar on a global scale. However that comparison is done using 0.5mg/ml of amotosalen and 0.5mg/ml of AMT- what happens when 10X more amotosalen is being used, as in the protocol? How does that look like at the global (comparing all pairwise crosslinks between transcriptomes) and in individual genes? Can the authors benchmark against true/false positives to understand the effect of extensive crosslinking?
- 3) The authors came up with an alternative way of extracting RNA besides using trizol. However, this is only important due to the extensive crosslinking of this sample. Are there advantages to using this way of RNA extraction, as compared to trizol, in other broader contexts? The authors need to show not only efficiency of extraction in wider contexts but also the intact levels of the RNA.
- 4) The authors used AO to limit UV damage during reverse crosslinking. However the experiments were done using a 8mer RNA fragment. This is very short compared to cleaved fragments in the transcriptome during library preparation. Can the authors show increased reversed transcription and decreased damage on a real transcript like actin B?
- 5) The authors claimed that with the same starting material, the library prep is improved by 76 fold, and that all of the steps combined resulted in a 4000X improvement. How is the 4000X calculated, what is the relative contribution of each improvement? Also, what is the amount of starting material used. If the library prep is improved by 76X, how low of a starting material can the authors start with for library prep? The authors need to show a clear example of a context whereby it would be beneficial to use PARIS2, instead of PARIS 1.
- 6) The authors claimed that PARIS2 acts as a powerful alternative method for direct analysis of mRNA translation. However, very little data is shown to support such a strong claim. Do the ribosome PARIS2 reads along mRNAs correlate with ribosome profiling data? Can the authors detect upstream open reading frames accurately? What exactly about mRNA translation can PARIS2 address specifically- that allows meaningful interpretation of translation status?
- 7) The authors showed U8 binding to U13 and hypothesized that this interaction is a molecular bridge that coordinates the biogenesis of rRNA subunits. The authors need to first validate this interaction using low throughput methods, and show that rRNA biogenesis is affected when U13 is knocked down.
- 8) The authors further showed the impact of SNVs on U8 structure using MFE modelling. The impact of these mutations of structure needs to be experimentally validated.
- 9) For enteroviruses, the authors showed the positive strand genome has much longer range of interactions than the negative strand genome. However the negative strand genome is way less abundant. Could the lack of long range interactions in the negative strand genome be due to technical

issues because of low abundance?

10) The authors should map their PARIS2 interactions onto known enterovirus structures to show that they can capture existing structures, and let the readers know how many false positives that they are capturing and how many interactions they are missing based on the existing structures.

11) The authors should also experimentally validate new interactions that are detected in enterovirus genomes.

12) Do mutations along any of the newly identified structures result in changes in virus fitness?

Reviewer #2 (Remarks to the Author):

This is a review for the manuscript titled, "Optimized photochemistry enables efficient analysis of dynamic RNA structures and interactomes in genetic and infectious diseases" The manuscript details the development and extension of a well-established method PARIS to capture RNA-RNA interactions. Being able to characterize RNA-RNA interactions is critical for our understanding of how RNA folds or forms intermolecular structures to contribute to biology. This manuscript extends the original method, PARIS, which relies on psoralen crosslinking to discover RNA-RNA interactions. PARIS2 (described herein) is applied to neurological disorder LCC, we discover a network of dynamic RNA structures and interactions which are destabilized by patient mutations. Also, the first genome structure of the RNA virus D68 is presented, revealing highly dynamic conformations altered by antiviral drugs and different pathogenic strains.

It is of the opinion of this reviewer that the methodological development is important, but the biological applications are not thorough and justified for publication in Nature Communications.

Overall, I believe this manuscript details key and systematic improvement over the PARIS method. However; I feel it is better suited for a more specialized journal in which the readership will utilize the important knowledge herein on PARIS library construction to move forward with utilizing PARIS for more biological studies.

Here are the points that need to be adequately addressed for publication.

The authors do not point out specific RNA structures which CANNOT be measured by PARIS1 that necessitate the development of PARIS2; therefore, the impact is hard to judge. And, also, because the new protocol has so many steps, it is unclear if it will be used by the community without clear demonstration that PARIS1 misses many key aspects of RNA structure that can be captured by PARIS 2.

I am very concerned about the authors use of chemotherapy RNA crosslinkers in their protocols. It is not clear how this is crosslinking RNA to induce crosslinks that are NOT present in the cells, and are instead, added after lysis in the lysate. The authors should present credible biochemical evidence that these known RNA crosslinking drugs are NOT inducing RNA crosslinks in their protocol, which may be false positives in sequencing analysis.

The authors present data for rRNA/mRNA crosslinking, but the arguments for the impact of this data are solely based on conjecture. To strengthen the utility of this approach, and to make stronger arguments for its utility to measure translation, the authors should compare mRNA crosslink sequences DIRECTLY to ribosome profiling. A prediction is that they should be correlated with higher PARIS2 crosslinks related to higher ribosome occupancy by ribosome profiling.

Figure 4 C is an empty box that does not have any data in it. That is the way it is presented in the PDF currently.

Supplementary Figure 24 is very difficult to read and interpret. I think it was altered during PDF compression. This should be fixed for subsequent submission and publication.

The work on viral RNA is very exciting and I think could really be used to better understand the importance of viral RNA and RNA-RNA interactions. However, I think the current study is missing key follow up experiments to directly test the importance of the interactions through viral infection assays or phenotypic screens to better understand RNA-RNA interactions and how they contribute to viral fitness. To put this into the context of the RNA structure field, there are other RNA structure manuscript recently that work to understand viral RNA structure with follow up experiments to characterize RNA elements directly. For example:

Structure mapping of dengue and Zika viruses reveals functional long-range interactions.
Huber RG, Lim XN, Ng WC, Sim AYL, Poh HX, Shen Y, Lim SY, Sundstrom KB, Sun X, Aw JG, Too HK, Boey PH, Wilm A, Chawla T, Choy MM, Jiang L, de Sessions PF, Loh XJ, Alonso S, Hibberd M, Nagarajan N, Ooi EE, Bond PJ, Sessions OM, Wan Y. *Nat Commun.* 2019 Mar 29;10(1):1408. doi: 10.1038/s41467-019-09391-8.

Integrative Analysis of Zika Virus Genome RNA Structure Reveals Critical Determinants of Viral Infectivity.
Li P, Wei Y, Mei M, Tang L, Sun L, Huang W, Zhou J, Zou C, Zhang S, Qin CF, Jiang T, Dai J, Tan X, Zhang QC. *Cell Host Microbe.* 2018 Dec 12;24(6):875-886.e5. doi: 10.1016/j.chom.2018.10.011. Epub 2018 Nov 21.

Reviewer #3 (Remarks to the Author):

This manuscript describes PARIS2, an improved method of profiling RNA structures and interaction in cells, which is based on the PARIS method previously developed partly by the senior author. PARIS2 incorporated several useful techniques such as applying a more water soluble version of psoralen and new RNA extraction methods. The authors claim a dramatic increase of efficiency by over 4000-fold. Although the techniques seem valid and the improvement is very impressive, it is not demonstrated to give meaningfully enhanced results. Ideally the new method would be compared with the old method for profiling of the same samples to show the improvement in the final results.

The authors went on to demonstrate three applications of PARIS2, which yielded a large amount of data of RNA structures and interactions. However, the study largely fell short of validating these interactions with alternative methods or functional experiments. Therefore, it would be difficult to access the degree of false positive in the results. For example, It would be helpful for the authors to compare the mRNA-ribosome binding data with previously described Riboseq data to cross validate the results. For another example, it would be helpful to perform mutation experiment of the virus genome sequence to validate the functional importance of detected interaction. These experiments would be very important for validation of the profiling results as well as the methodology.

On the format, the writing and organization of the manuscript made it a bit difficult to read. Some of the technical descriptions in the main text could be transferred to supplementary materials to make the main text more friendly to general readership. The figures, especially Figure 5 and 6 are somewhat too crowded. Some trimming or reorganization would be helpful.

Summary of revision in response to reviewers' comments.

We sincerely appreciate the reviewers' detailed and constructive comments, which have greatly improved the manuscript. In response, we have substantially revised the manuscript. Our answers below and revised text in the manuscript are highlighted in blue. Clarifications were made in cases where the original submission already included necessary evidence in the extensive supplementary figures, including the importance of the new amotosalen crosslinker (**S. Fig. 2**), evidence for singlet quencher protection of RNA (**Fig. 2n and S. Fig. 10**), calculated fold improvements for each step of PARIS2 (**S. Tables 1-2**), validation of PARIS2 on known viral structures (**S. Figs. 25-27, 30, 31**), confidence in the newly discovered viral RNA structures (**S. Figs. 24, 26, 27, 31**), and chemotherapy RNA crosslinkers (**S. Figs. 4-5**),

More importantly, new validation and functional analysis data are incorporated into Supplementary Figures, including detailed comparison between PARIS1 and PARIS2, and amotosalen at 0.5 or 5mg/ml (**S. Fig. 2e-m**), systematic comparison between PARIS2 and Ribo-Seq (**S. Fig. 19i-m**), validation of U8-U13 interaction (**S. Fig. 20m-n**), validation of the destabilizing effects of LCC patient derived mutations on U8 structures and interactions (**S. Fig. 21h-i**), and mutational analysis of the role of a newly discovered alternative conformation in the EV-D68 IRES (**S. Fig. 32**).

Reviewer #1 (Remarks to the Author):

The manuscript from Zhang M et al optimized several steps of the PARIS protocol to enable better capture of RNA-RNA interactions genome-wide. They then applied it to a variety of questions including translation, U8 interactions and the RNA interactome of enteroviruses. While a tremendous amount of work has been done, the manuscript comes across as scattered, with each sub-story being poorly developed. The authors may benefit from splitting this manuscript into better developed individual manuscripts.

We appreciate the reviewer's recognition of the dramatic improvements. Despite the long history of chemical crosslinking in RNA structure and interaction studies, surprisingly little is known about almost every single step in this sophisticated strategy, and it is necessary to perform the systematic investigations that span multiple disciplines, including chemistry, enzymology, and even techniques that are as basic as electrophoresis (e.g. the DD2D gels). Therefore, we believe splitting the manuscript is not a good choice, as all the technological innovations are required for the high efficiency, and essential to move the field forward.

We were fully aware of the difficulties in reading such a dense technology paper, which is why we wrote 5 lengthy **Supplementary Notes** (much longer than the main text), each like a "mini-paper", describing in detail the historical background of each problem, mechanistic studies of the fundamental chemical/physical properties, systematic screening for optimal conditions, and extensive validation and comparison with conventional approaches. For example, this reviewer is interested in seeing broader applications of the TNA method (see below for detailed response to point 3), which we have shown on different chemical crosslinkers (psoralens, and chemotherapy drugs BCNU and CHL), and the enrichment of mRNAs, snoRNAs and viral RNAs, all of which requires intact RNAs for antisense enrichment. In other words, the diverse applications strengthened the claims of the broad utility of PARIS2.

In fact, in the past 4 years, we have been frequently approached by colleagues for help with specific applications of PARIS, most of which focus on specific, low-abundance RNAs in challenging samples and disease contexts. This systematic overhaul of PARIS has finally made it possible for these applications. We acknowledge that demonstrating broad applications made it difficult to dig deeper into each particular case. In this revision, we are providing further validations and functional analysis for the biological applications, including the ribosome-mRNA interactions, the snoRNA network and viral RNA structures, which hopefully addresses the concerns raised above. See responses below for the details of these revisions.

Here are the major comments:

1) The authors developed amotosalen because they say that AMT is only soluble at 1 mg/ml. While efforts to develop a new psoralen compound is admirable, this reviewer is unsure why there is a need for this new compound in this protocol. AMT's solubility is at 1mg/ml. That is already highly soluble. Crosslinking experiments done at 0.5mg/ml inside cells are already so extensive that RNA is being crosslinked to proteins, resulting in a need to use proteinase K during RNA extraction. As such, more crosslinking is not necessarily better. There is a possibility of everything will be crosslinked to everything, making it difficult to determine which crosslinks are necessarily biologically relevant. Can the authors show the benefits of having even more crosslinking using amotosalen? Can they capture additional real crosslinks that were missing in PARIS? How many more real crosslinks are they capturing? What is the false positive rate?

We thank the reviewer for recognizing the improvement in photochemistry. The motivation behind developing amotosalen was the low efficiency of AMT at 0.5mg/ml. In our previous study (Lu et al. 2016 Cell, and similar studies from other groups), we were only able to obtain 0.5% crosslinked RNA fragments, far lower than expected. For example, rRNAs make up ~70% mass of the transcriptome, in which double-stranded regions constitute >70% of the sequence, and ~50% of the duplexes have the potential crosslinking sites (staggered U-U, U-C and C-U pairs). Given these estimates, we expect at least 25% RNA fragments to be crosslinkable. By using the higher solubility amotosalen at 5mg/ml, compared to AMT at 0.5mg/ml, we were able to improve crosslinking efficiency by ~7 fold, to ~4.5% (**Supplementary Figure 2b** and **Supplementary Table 1**). Even though this is still much lower than the theoretical limit of 25%, this is still very remarkable, given that AMT has been the working horse for 40 years in the field, and little progress has been made since its first synthesis and application in 1977 (Issacs et al. 1977 Biochemistry).

The 1mg/ml solubility for AMT, ~3mM, is not very high. In comparison, SHAPE reagents, e.g. NAI-N3, are routinely used at 100mM (Spitale et al. 2015 Nature), while DMS (dimethyl sulfide) are often used at 200-400mM (Rouskin et al. 2014, Nature). The use of proteinase K is necessary regardless of psoralen concentration, since psoralens have the natural ability to crosslink nucleic acids to proteins, even though with lower efficiency than between nucleic acids (Sastri et al. 1997, PMID: 9188273, and for example in **Supplementary Fig. 3a**, 20ug/ml already led to significant RNA-protein crosslinking). The need for proteinase K is therefore NOT an indicator of spurious non-specific crosslinking. In fact, the proteinase K treatment is highly efficient, degrading most proteins down to 2-6 amino acids, therefore, there is little concern about random protein-mediated RNA interactions. On the other hand, given that crosslinking reduces molecule mobility (like fixing cells with formaldehyde), there is little concern about structural rearrangements during the extensive crosslinking.

In order to further evaluate the sensitivity and specificity of PARIS2, we performed ROC analysis using the cryo-EM structure of the 28S rRNA as a gold standard (PDB: 4V6X). The base pairing status in every 5-nt bin in 28S rRNA was calculated based on cryo-EM and PARIS2 data. PARIS1 and PARIS2 have similar performance (**Supplementary Fig. 2i**, copied on the right). Different psoralen crosslinkers at different concentrations also perform similarly (**Supplementary Fig. 2j**). In summary, the newly synthesized amotosalen dramatically improved crosslinking efficiency due to its higher solubility, without causing more artifacts.

2) Similar to point 1 above. The authors claim that the RNA duplexes captured by AMT and amotosalen are similar on a global scale. However, that comparison is done using 0.5mg/ml of amotosalen and 0.5mg/ml of AMT- what happens when 10X more amotosalen is being used, as in the protocol? How does that look like at the global (comparing all pairwise crosslinks between transcriptomes) and in individual genes? Can the authors benchmark against true/false positives to understand the effect of extensive crosslinking?

We thank the reviewer for this great suggestion. We have now included more PARIS2 data on 5mg/ml amotosalen crosslinked cells and compared with the sequencing data from cells crosslinked with 0.5mg/ml AMT, and 0.5mg/ml amotosalen. On the global level, the total gapped+chimeric reads ratio (**Supplementary Fig. 2e**, copied on the right) and the identified RNA types (**Supplementary Fig. 2f**) are similar among 0.5mg/ml AMT, 0.5mg/ml and 5.0mg/ml amotosalen crosslinked samples. To compare the intra-molecular structures identified using the 3 conditions, we assembled gapped reads to duplex groups (DGs). Comparable numbers of DGs in the 18S and 28S rRNA were identified (**Supplementary Fig. 2g**). Inter-molecular interactions among rRNAs were identified using chimeric reads and they are also similar among different conditions (**Supplementary Fig. 2h**). The lack of rRNA-mtrRNA (cytoplasmic vs. mitochondrial rRNAs) interactions further suggested that 5.0mg/ml amotosalen did not cause additional artifacts (**Supplementary Fig. 2h**). See response above to point 1 for the analysis of accuracy and specificity in detecting 28S rRNA structures (**Supplementary Fig. 2i-j**).

In addition, we compared U2, U4 and U6 snRNA internal structures and interactions. Results showed that AMT and amotosalen (both 0.5 and 5.0 mg/ml) were similar in detecting RNA structure and interactions (**Supplementary Fig. 2k-m**, copied below). There are only some minor differences in the intensity of specific DGs, reflecting the different strengths of crosslinking. Such differences do not affect their applications, since comparisons among different biological conditions will be performed using identical

PARIS2 protocols (i.e. the same crosslinker and concentration).

3) The authors came up with an alternative way of extracting RNA besides using trizol. However, this is only important due to the extensive crosslinking of this sample. Are there advantages to using this way of RNA extraction, as compared to trizol, in other broader contexts? The authors need to show not only efficiency of extraction in wider contexts but also the intact levels of the RNA.

We apologize for the confusion in the description, for which we provide further clarification here. TNA is not an alternative, but the **ONLY** method that extracts all crosslinked RNA from cells. For the purpose of complete recovery, it is absolutely necessary even for mildly crosslinked samples, not just extensively crosslinked ones. As shown in **Supplementary Fig. 3a**, even 20ug/ml AMT crosslinking caused RNA loss during the conventional TRIZol extraction. For heavier crosslinked samples, the advantage of using TNA is much bigger (**Supplementary Fig. 6**), when compared to gold standard methods, such as TRIZol and RNeasy (>6 fold when applied to 5mg/ml crosslinked RNA).

Besides AMT and amotosalen, we showed that TNA is necessary to extract RNA from other types of samples, e.g. ones crosslinked with chemotherapy drugs carmustine (BCNU) and chlorambucil (CHL) (**Supplementary Figures. 4-5**), demonstrating its broad applications in the studies of RNA structures and interactions. It is likely that even more types of crosslinkers also induce higher hydrophobicity in RNA and abnormal phase partition, and our work revealed an important caveat in the applications of TRIZol that should be considered in ANY similar studies in the future. In fact, we were extremely surprised by this discovery, given that psoralen crosslinking has been used for more than 40 years, yet no one has noticed this problem. As such, earlier studies using psoralens should also be interpreted with caution.

In another study, it was shown that structured microRNAs with low GC content are selectively lost during TRIZol extraction from a small number of cells, further demonstrating the potential problems associated with improper use of the gold standard AGPC method (e.g. TRIZol, Kim et al. 2012, PMID: 22749402).

The integrity of TNA-extracted RNA samples was already demonstrated in multiple places in the original submission (**Fig. 2j**, **Supplementary Figs. 4 and 6**). Based on Bioanalyzer and TapeStation analysis, the RIN number is typically >9 for non-crosslinked RNA extracted using TNA. Strong crosslinking causes broad smear across the range of size distribution, so the RIN numbers are no longer an accurate indicator of intactness. In **Supplementary Fig. 5**, BCNU crosslinking itself causes RNA degradation, so it is not TNA's failure to recover intact RNA. Nevertheless, TNA method recovers all RNA with high integrity.

4) The authors used AO to limit UV damage during reverse crosslinking. However the experiments were done using a 8mer RNA fragment. This is very short compared to cleaved fragments in the transcriptome during library preparation. Can the authors show increased reversed transcription and decreased damage on a real transcript like actin B?

The suggested experiments were already included in the original submission. We performed RT-qPCR to test the AO protection of RNA against UVC damage. The same amount of cellular RNA was irradiated by 254nm UV with or without AO protection. After 10 or 30 min irradiation, reverse transcribed cDNA from ACTB and GAPDH mRNAs were measured using qPCR (amplicon size at 180 and 70 nt, respectively, **Figure 2n** and **Supplementary Figure 10c-f**). Results suggested that AO effectively protects non-crosslinked RNA even after 30min UVC irradiation, after which 30%

RNA remain intact, vs. 0.5% in the absence of AO (**Figure 2n, upper panel**). AO also protects psoralen crosslinked RNA from UVC damage (**Figure 2n, lower panel**).

5) The authors claimed that with the same starting material, the library prep is improved by 76 fold, and that all of the steps combined resulted in a 4000X improvement. How is the 4000X calculated, what is the relative contribution of each improvement? Also, what is the amount of starting material used. If the library prep is improved by 76X, how low of a starting material can the authors start with for library prep? The authors need to show a clear example of a context whereby it would be beneficial to use PARIS2, instead of PARIS 1.

We calculated the improvements of sample preparation and library preparation separately in the original submission, and the details were already presented in **Supplementary Tables 1-2**. Here we further clarify the calculations.

Sample preparation includes 3 steps: crosslinking, RNA extraction and DD2D gel purification. In PARIS2, 5.0mg/ml amotosalen resulted in 6.95-fold more crosslinked RNA comparing to 0.5mg/ml AMT in PARIS1 (**Supplementary Tables 1-2**); TNA retrieved 6.48-fold crosslinked RNA than TRIZol; DD2D gel purification increased recovery 1.54-fold comparing to ND2D in PARIS1. The total improvement of sample preparation is $6.95 * 6.48 * 1.54 = 69.35$, which means that we recovered 69.35-fold more crosslinked RNA fragments from the same number of cells.

Library preparation includes 4 improvements: adapter ligation, UVC protection (AO), PUVA bypass (SSIV) and PUVA bypass (Mn^{2+} buffer). The total library yield improvement of 75.75-fold was based on results of library yield from the same amount of starting crosslinked RNA fragments after 2D gel. This number is close to the lower bound of the multiplication of fold improvements for these library preparation steps, which is in the range of [131, 2956] (**Supplementary Table 1**). This is because the several steps could be bottlenecks at the same time and improving one step may not be sufficient to lift the yield for the entire pipeline. Together, these optimizations in PARIS2 resulted in >4000-fold ($69.35 * 75.75$) increased efficiency.

In addition to the increased efficiency, the TNA method made it possible to perform antisense-mediated enrichment of specific transcripts, which was either impossible before (PARIS1), or was highly inefficient because most crosslinked RNAs were lost during typical TRIZol and RNeasy extractions (e.g. SPLASH, LIGR-seq and COMRADES). For example, in this manuscript we demonstrated the efficient recovery of mRNAs, snoRNAs and viral RNAs using the targeted antisense enrichment method, which was not possible before using PARIS1.

6) The authors claimed that PARIS2 acts as a powerful alternative method for direct analysis of mRNA translation. However, very little data is shown to support such a strong claim. Do the ribosome PARIS2 reads along mRNAs correlate with ribosome profiling data? Can the authors detect upstream open reading frames accurately? What exactly about mRNA translation can PARIS2 address specifically- that allows meaningful interpretation of translation status?

We thank the reviewer for this great suggestion. Here we provide additional analysis of mRNA-rRNA interactions from PARIS2 data. To compare PARIS2 with ribosome profiling (Ribo-Seq), we used published Ribo-Seq data from HEK293 cells (SRR1630831, Gao *et al.* Nat Methods 2015). We found that ribosome binding on mRNA CDS, 5'UTR and 3'UTR were comparable between PARIS2 and Ribo-Seq (**Supplementary Fig. 19i**, copied here). The minor differences in coverage are likely due to dramatic differences in the methodology. mRNAs with ribosome binding level (RPKM ≥ 10) were analyzed here (PARIS2: 0.47 million of 18S rRNA chimeric alignments; Ribo-Seq, 4.5 million alignments). Among transcripts detected by both PARIS2 and Ribo-Seq, 2144 of them are common (**Supplementary Fig. 19j**, Fisher's exact test $p = 5.12 \times 10^{-257}$), and there is strong positive correlation between them in coverage (**Supplementary Fig. 19k**).

Ribosome binding profile around translation start sites on meta mRNAs were also examined. We found that h26 binding site precedes that of h18, consistent with their locations in the mRNA channel. Ribo-Seq profile is more similar to h26 binding profile (**Supplementary Fig.**

19l, with the maximum normalized to 1). We also showed good correlation between PARIS2 and Ribo-Seq in four example mRNAs (**Supplementary Fig. 19m**, with the maximum normalized to 1). We were unable to analyze uORFs at the moment due to low sequence coverage from current PARIS2 libraries, since most well characterized uORFs are located in low-abundance mRNAs.

7) The authors showed U8 binding to U13 and hypothesized that this interaction is a molecular bridge that coordinates the biogenesis of rRNA subunits. The authors need to first validate this interaction using low throughput methods, and show that rRNA biogenesis is affected when U13 is knocked down.

We thank the reviewer for this great suggestion. We have used RNA antisense purification combined with qPCR to verify the U8-U13 interaction (**Supplementary Fig. 20m-n**, copied here). Briefly, after crosslinking cells and purifying total RNA, U8 antisense oligos (ASOs) were added to cellular RNA to pull down U8 and its associated RNAs. Reverse crosslinking and RT-qPCR were then performed to test enrichment of targets. U13 was enriched by U8 pull down from both mouse brain and human neuroprogenitor cells (NPC), while two other RNAs, *ACTB* mRNA and U1 snRNA were not enriched (using species-specific primers).

Early studies have already shown that snoRNA U13 was required for pre-rRNA processing through base-pairing to the 3' end of 18S rRNA (Cavaillé *J et al.*, *Eur J Biochem.* 1996). Our PARIS2 data confirmed that U13 bind to 3' end of 18S rRNA (**Figure 4a**). Sharma *et al.* found that U13 was essential for acetylation deposition of C₁₃₃₇ of the 18S rRNA in humans and C₁₂₉₇ of the 18S rRNA in yeast (Sharma *et al.*, *Nucleic Acids Res.* 2015). And this acetylation modification was involved in 18S rRNA biogenesis. However, the function of the specific interaction between U8 and U13 remains to be determined. Ongoing studies in our lab are investigating the involvement of this network of interactions in ribosome biogenesis and the neurodegenerative disease LCC.

8) The authors further showed the impact of SNVs on U8 structure using MFE modelling. The impact of these mutations of structure needs to be experimentally validated.

We thank the reviewer thank this great suggestion. We have now validated the destabilizing effect of one LCC-patient derived U8 mutation, 3G>A, on RNA structure. We synthesized 22nt fragments on the 5' end of the wildtype (WT) and 3G>A mutant (Mut) U8 snoRNA. The 3G>A mutation dramatically reduces dimer and internal hairpin stability based on MFE predictions (**Supplementary Fig. 21h**, predicted secondary structures and MFE, copied here). We measured the melting curve of these two oligos and observed a dramatic reduction in melting temperature, from 65 to 54°C (**Supplementary Fig. 21h**, fluorescence measurements at increasing temperatures), primarily reflecting the destabilization of the dimer forms since the intra-molecular hairpin is not stable at all at these higher temperatures.

9) For enteroviruses, the authors showed the positive strand genome has much longer range of interactions than the negative strand genome. However, the negative strand genome is way less abundant. Could the lack of long range interactions in the negative strand genome be due to technical issues because of low abundance?

The lack of detection of the long-range structures is not due to lower abundance, since our analysis already normalized the abundance between the two strands. In fact, we were able to detect the long-range structures on the positive strand even after sub-sampling reads to the same coverage of the negative strand.

10) The authors should map their PARIS2 interactions onto known enterovirus structures to show that they can capture existing structures, and let the readers know how many false positives that they are capturing and how many interactions they are missing based on the existing structures.

To benchmark the sensitivity of PARIS2, we already presented extensive data on the identification of previously reported structures in the original submission, including 5'cloverleaf (5'-CL) and internal ribosome entry site (IRES) located at 5'UTR, 2C-CRE located at coding region, and 3'UTR stemloops (**Supplementary Figs 25-27, 30 and 31**).

All of them are captured, giving a 100% true positive rate. It is almost impossible to determine false positives in the enterovirus genomes, because no in vivo genome-wide structure models have been published before. Such analysis has been performed using well studied cellular RNAs, such as rRNAs, snRNAs and many others (see Lu et al. 2016 Cell, and response above on rRNAs and snRNAs, points 1-2, and details in **Supplementary Fig. 2**). In fact, even these types of analysis are still an over-estimation of false positives. Cellular RNAs almost always fold into more dynamic and alternative conformations than can be captured by in vitro methods, such as crystallography and cryo-EM, as a result no one truly knows all the possible conformations.

11) The authors should also experimentally validate new interactions that are detected in enterovirus genomes.

We have already presented extensive data that support the validity of newly discovered RNA structures. As described above, we performed extensive benchmarking studies on the technology (**Supplementary Fig. 2**). We showed that the PARIS2 and PARIS1 are similar in their accuracy and specificity (see response to points 1-2), and the original PARIS method was also extensively validated (Lu et al. 2016 Cell). We validated multiple types of structures and interactions in well-studied rRNAs and snRNAs (**Supplementary Fig. 2**). In this manuscript, we also provided new evidence validating newly discovered structures and interactions in the U8 snoRNA network (**Supplementary Figs. 20-21**). All these studies lend credibility to the PARIS2 method and new discoveries.

We also used multiple different approaches to ensure the reproducibility and validity of newly discovered enteroviral genome structures. First, we performed PARIS2 using multiple different conditions, including different strains, host cell types, inhibitor treatments (**Supplementary Fig. 24**). In addition to identifying known structures and condition-specific structures, our comparison among these conditions confirmed the reproducibility (especially in **Supplementary Fig. 24c**, US47 strain in two cell lines). Second, we analyzed the conservation of a number of newly discovered structures and found high concordance between PARIS2 and genome-alignment derived models (**Supplementary Figs. 26, 27 and 31**).

Finally, we would like to point out that in vivo crosslinking, like PARIS, is a unique class of methods that cannot be easily validated using in vitro approaches that lack the cellular context critical for the formation and dynamic arrangements of structures. In vivo crosslinking and qRT-PCR is basically the same as crosslinking plus sequencing, which we already showed for the U8-U13 interaction. These technologies are similar to chromatin conformation captures, where in vivo conformations are extremely difficult to recapitulate in vitro. Results from in vitro prepared RNA fragments, either positive or negative, cannot be used to conclusively validate or refute in vivo structures. See response below for functional validation of one newly discovered alternative conformation.

12) Do mutations along any of the newly identified structures result in changes in virus fitness?

We thank the reviewer for this great suggestion. To determine whether alternative structures identified by PARIS2 are functional, we designed mutations for domain IV and the newly identified alternative structure (a3) to disrupt their structures, and tested their effects on IRES activity (**Supplementary Fig. 32a**). For domain IV, 13 mutations were introduced to disrupt the structure. For a3, we constructed mutants that disrupted base pairings (a3-mut1: U446C+C454G or a3-mut2: C447G+C454G). In addition, compensatory mutations were introduced to a3-mut1 and a3-mut2 to restore the base pairings (a3-res1: U446C+ C454G/A424G+G416C or a3-res2: C447G+C454G/G423C+G416C) (**Supplementary Fig. 32b, d**). These compensatory mutations may also affect the structure of domain IV as predicted, so another mutation (C266G) was introduced to maintain the RNA model of domain IV as much as possible. All EVD68 IRES fragments, including wildtype, mutated and rescued ones, were individually inserted between the Renilla and firefly luciferase genes (**Figure R6A**). Firefly and Renilla luciferase activities were measured and IRES activity was calculated as the ratio of Firefly luciferase activity to Renilla luciferase activity (**Supplementary Fig. 32c, e**).

First, compared to wildtype IRES, both IV-mut and a3-mut (a3-mut1 or a3-mut2) showed significantly decreased IRES activities, indicating they are necessary for IRES. Second, the remarkably greater decrease of IRES activity of the double mutant IV-mut/a3-mut (a3-mut1 or a3-mut2) further supports this conclusion because if either one is nonfunctional, the IRES activity of the IV-mut/a3-mut will only remain the same level as that of single mutants. Here we were unable to restore the IRES activity by rescuing a3 structure (a3-res1 or a3-res2), and as a result cannot completely exclude the possibility that the single-stranded conformation (i.e. sequence alone) is necessary for IRES activity. We speculated that the incomplete restoration of domain IV structure may still have a negative effect on IRES activity, which results in similar level of a3-res activity to that of a3-mut. Given the high frequency of the a3 conformation (**Fig. 6a**), these experiments suggest that the a3 is required for EVD68 IRES activity.

Supplementary Figure 32. Alternative structures play important roles in EV-D68 IRES activity. a, Schematic of bicistronic luciferase reporter plasmids. Various EV-D68 IRES fragments, including Wildtype, three mutants and compensatory rescue were inserted between the Renilla and firefly luciferase genes. The negative control in this assay has no insertion between these two genes. Renilla luciferase activity was measured to detect cap-dependent translation, while firefly luciferase activity was measured to detect IRES-mediated translation. IRES activity was calculated as the ratio of firefly luciferase activity to Renilla luciferase activity. b,d, Predicted secondary structure models of domain IV and alternative structure 3 (a3), including IV-mut only, a3-mut only, IV-mut + a3-mut, a3-res and WT (wildtype). The green arrows point to positions of mutations. The black arrows point to compensatory rescue mutations. The MFE values were shown below the respective models. c,e, Effects of domain IV and a3 mutations on IRES activity. HEK 293T cells were transfected with pR/F-IRES plasmids, and 24 h later, firefly and Renilla luciferase activities were individually measured, and IRES activity was calculated as the ratio of firefly luciferase activity to Renilla luciferase activity. All data are representative of at least three independent experiments. The results are presented as the mean \pm SEM (* $p < 0.05$, ** $p < 0.01$, and *** $p < 0.001$).

Reviewer #2 (Remarks to the Author):

This is a review for the manuscript titled, "Optimized photochemistry enables efficient analysis of dynamic RNA structures and interactomes in genetic and infectious diseases" The manuscript details the development and extension of a well-established method PARIS to capture RNA-RNA interactions. Being able to characterize RNA-RNA interactions is critical for our understanding of how RNA folds or forms intermolecular structures to contribute to biology. This manuscript extends the original method, PARIS, which relies on psoralen crosslinking to discover RNA-RNA interactions. PARIS2 (described herein) is applied to neurological disorder LCC, we discover a network of dynamic RNA structures and interactions which are destabilized by patient mutations. Also, the first genome structure of the RNA virus D68 is presented, revealing highly dynamic conformations altered by antiviral drugs and different pathogenic strains.

1. It is of the opinion of this reviewer that the methodological development is important, but the biological applications are not thorough and justified for publication in Nature Communications. Overall, I believe this manuscript details key and systematic improvement over the PARIS method. However; I feel it is better suited for a more specialized journal in which the readership will utilize the important knowledge herein on PARIS library construction to move forward with utilizing PARIS for more biological studies.

We appreciate the reviewer's recognition of the significant improvement that we report in this paper. We have now provided more validations and functional analysis that strengthened and broadened its impact, including further data on mRNA-rRNA interactions, U8 interaction network, validation of LCC patient mutations on destabilization of U8 structures, and functional analysis of alternative structures in virus genomes (see responses above and below).

In addition, we also want to point out that the technical improvements are not "simple optimizations", and certainly not just for PARIS. These deep mechanistic insights came from several years of exhaustive studies into the fundamental aspects of RNA chemistry, photochemistry, enzymology, and have broad impacts in several different fields (compressed in **Fig. 2** due to space limit, and presented in detail in 15 **Supplementary Figures** and 5 extensive **Supplementary Notes**). For example, the high efficiency amotosalen will be useful for both RNA and DNA crosslinking studies (including DNA torsional stress, Teves and Henikoff 2013, <https://www.nature.com/articles/nsmb.2723>, chromosome conformation analysis, <https://www.nature.com/articles/s41587-020-0643-8>). The discovery of crosslinking-induced hydrophobicity and subsequent development of TNA is absolutely essential for the future applications of crosslinking in the analysis of RNA structures, especially given its increasing popularity in recent years. The ability to prevent and bypass photochemical damages in RNA is important for many types of RNA research given the prevalent use of UV, e.g. RNA-protein crosslinking, UV-based imaging and quantification of RNA (Kladwang et al. 2012, <https://www.nature.com/articles/srep00517>), etc. Together, our study solved multiple decades-old problems in RNA research. Therefore, we respectfully request that these significant advancements not be overlooked.

2. Here are the points that need to be adequately addressed for publication. The authors do not point out specific RNA structures which CANNOT be measured by PARIS1 that necessitate the development of PARIS2; therefore, the impact is hard to judge. And, also, because the new protocol has so many steps, it is unclear if it will be used by the community without clear demonstration that PARIS1 misses many key aspects of RNA structure that can be captured by PARIS 2.

The motivation for PARIS2 is the overall low efficiency of PARIS and all other similar methods, such as SPLASH, LIGR-seq and COMRADES. This low efficiency is the result of several fundamental problems in the fields of photochemistry, RNA chemistry and enzymology, which we have finally solved in this manuscript, including: **(1)** Low solubility and low efficiency of AMT. As a result, most RNAs are not crosslinked (see response to reviewer 1, point 1). **(2)** Failure to extract crosslinked RNA by the widely used and classical AGPC (e.g. TRIzol) and column-based purification (e.g. RNeasy) methods. As a result, most crosslinked long RNAs are selectively lost (**Fig. 2b-c** and **Supplementary Fig. 6**). **(3)** Extensive PUVA and UVC induced RNA damages that reduced reverse transcription efficiency. These bottlenecks affect the detection of every RNA structure, although not equally.

Structures that "CANNOT be measured by PARIS1" include ones that do not have staggered uridines, due to the inherent chemical bias of psoralens. We acknowledge that the amotosalen introduced in this manuscript, despite its much higher efficiency, still did not solve this problem. We are currently developing new chemical crosslinkers that have no bias towards the different bases. Despite this limit, we believe that the broad applicability of the new methods in RNA research, >4000-fold increased efficiency and enabling efficient targeted enrichment for the first time (mRNAs, snoRNAs, viral RNA genomes) are critical for future applications of the crosslink-ligation method.

The new PARIS2 does NOT have more steps compared to the original. The basic principle and pipeline remain the same, with dramatic revisions to each step. To facilitate broad application, we wrote a very detailed protocol, including indicators of success or failure for every step, and trouble-shooting tips for critical steps (see the **Supplementary Protocol**). In addition, the **5 Supplementary Notes** (11 pages plus 80 references) provided a comprehensive overview of the background information for all major technical issues in crosslinking-ligation based technologies, rationale for optimizations, mechanistic studies and systematic screens for optimal conditions. These notes, each as a "mini-paper", complements the main text and the protocol, serving as a solid foundation for the application of PARIS2, and the entire field of RNA photochemistry.

3. I am very concerned about the authors use of chemotherapy RNA crosslinkers in their protocols. It is not clear how this is crosslinking RNA to induce crosslinks that are NOT present in the cells, and are instead, added after lysis in the lysate. The authors should present credible biochemical evidence that these known RNA crosslinking drugs are NOT inducing RNA crosslinks in their protocol, which may be false positives in sequencing analysis.

We apologize for the lack of clarity in the description. We did NOT use the chemotherapy drugs for PARIS2 experiments due to their lack of reversibility. We performed the BCNU and CHL crosslinking experiments primarily to demonstrate the generality of the crosslinking-induced hydrophobicity, the necessity of the TNA purification method and the general applicability of the DD2D gel system (**Supplementary Figs. 4-5, Supplementary Notes 1 and 3**).

Together with AMT and amotosalen, results from these crosslinkers strengthened our conclusions on the improvements.

4. The authors present data for rRNA/mRNA crosslinking, but the arguments for the impact of this data are solely based on conjecture. To strengthen the utility of this approach, and to make stronger arguments for its utility to measure translation, the authors should compare mRNA crosslink sequences DIRECTLY to ribosome profiling. A prediction is that they should be correlated with higher PARIS2 crosslinks related to higher ribosome occupancy by ribosome profiling.

We thank the reviewers for this great suggestion. We have now directly compared PARIS captured rRNA-mRNA interactions with ribosome profiling data in the same cell type HEK293 (**Supplementary Fig. 19i-m**, see response to reviewer 1, point 6).

5. Figure 4 C is an empty box that does not have any data in it. That is the way it is presented in the PDF currently. Supplementary Figure 24 is very difficult to read and interpret. I think it was altered during PDF compression. This should be fixed for subsequent submission and publication.

We are sorry for this mistake. We have now fixed this problem.

6. The work on viral RNA is very exciting and I think could really be used to better understand the importance of viral RNA and RNA-RNA interactions. However, I think the current study is missing key follow up experiments to directly test the importance of the interactions through viral infection assays or phenotypic screens to better understand RNA-RNA interactions and how they contribute to viral fitness. To put this into the context of the RNA structure field, there are other RNA structure manuscript recently that work to understand viral RNA structure with follow up experiments to characterize RNA elements directly. For example:

Structure mapping of dengue and Zika viruses reveals functional long-range interactions. Huber RG, Lim XN, Ng WC, Sim AYL, Poh HX, Shen Y, Lim SY, Sundstrom KB, Sun X, Aw JG, Too HK, Boey PH, Wilm A, Chawla T, Choy MM, Jiang L, de Sessions PF, Loh XJ, Alonso S, Hibberd M, Nagarajan N, Ooi EE, Bond PJ, Sessions OM, Wan Y. Nat Commun. 2019 Mar 29;10(1):1408. doi: 10.1038/s41467-019-09391-8.

Integrative Analysis of Zika Virus Genome RNA Structure Reveals Critical Determinants of Viral Infectivity. Li P, Wei Y, Mei M, Tang L, Sun L, Huang W, Zhou J, Zou C, Zhang S, Qin CF, Jiang T, Dai J, Tan X, Zhang QC. Cell Host Microbe. 2018 Dec 12;24(6):875-886.e5. doi: 10.1016/j.chom.2018.10.011. Epub 2018 Nov 21.

We appreciate the reviewer's recognition of the importance of in vivo structure analysis on RNA viruses and agree that our mechanistic studies of viral RNAs remain preliminary. As requested, we have now performed functional tests on a newly discovered alternative structure in the EVD68 IRES (**Supplementary Fig. 32**, see detailed response to reviewer 1, point 12).

Our coverage of several different topics in the applications, such as rRNA-mRNA interactions, snoRNA networks in a neurodegenerative disease, and viral genomes certainly diluted the focus, whereas the referenced papers above solely focused on applying established methods to viral structures and functions. As such, it is not a fair comparison.

We also want to point out that the broad and deep mechanistic studies of photochemistry, RNA chemistry and enzymology, and the systematic screening for optimal conditions (unjustly compressed in **Fig. 2**, but detailed in 5 comprehensive **Supplementary Notes**, each like a "mini-paper", and 15 **Supplementary Figures**), is unparalleled in the entire RNA structure field, and certainly not in the two papers listed above. In fact, the Li et al. work came from our collaborator and used our PARIS method to build the first Zika genome structures. Much deeper sequencing was performed in these two studies due to the lack of a good enrichment strategies. We respectfully request that these significant mechanistic and technological advancements not be overlooked when assessing the overall impact of our study.

Reviewer #3 (Remarks to the Author):

1. This manuscript describes PARIS2, an improved method of profiling RNA structures and interaction in cells, which is based on the PARIS method previously developed partly by the senior author. PARIS2 incorporated several useful techniques such as applying a more water-soluble version of psoralen and new RNA extraction methods. The authors claim a dramatic increase of efficiency by over 4000-fold. Although the techniques seem valid and the improvement is very impressive, it is not demonstrated to give meaningfully enhanced results. Ideally the new method would be compared with the old method for profiling of the same samples to show the improvement in the final results.

We sincerely thank the reviewer for recognizing the dramatic improvements, which really came from several years of deep mechanistic studies into photochemistry, RNA chemistry, enzymology, electrophoresis and computational biology. We were extremely surprised during technology development that despite decades of work on psoralens (particularly AMT), so little was known about the fundamental mechanisms, which limited its efficiency and biological applications. In the early days, the application of AMT was critical for establishing the mechanisms of splicing by the abundant snRNAs, largely in the labs of Steitz, Pederson, and Luhrmann. Lifting these long-standing bottlenecks will further facilitate broad applications. Here we present evidence, including new data, demonstrating the benefits of the improvements. We especially want to highlight the broad impact of these mechanistic insights in RNA research. These improvements are valuable in their own rights, not just for the purpose of improving PARIS2 (see the 5 **Supplementary Notes** for details).

(1). The motivation behind the development of better crosslinkers was the low efficiency of AMT at 0.5mg/ml. In our previous study (Lu et al. 2016 Cell, and similar studies from other groups), we were only able to obtain 0.5% crosslinked RNA fragments, far lower than expected. For example, rRNAs make up ~70% mass of the transcriptome, in which double-stranded regions constitute >70% of the sequence, and ~50% of the duplexes have the potential crosslinking sites (staggered U-U, U-C and C-U pairs). Given these estimates, we expect at least 25% RNA fragments to be double stranded and crosslinkable. By using the higher solubility amotosalen at 5mg/ml, compared to AMT at 0.5mg/ml, we were able to improve crosslinking efficiency by ~7 fold, to ~4.5% (**Supplementary Figure 2b** and **Supplementary Table 1**). Even though this is still much lower than the theoretical limit of 25%, this is still very remarkable, given that AMT has been the working horse for 40 years in the field, and little progress has been made since its first synthesis in 1977 (Issacs et al. 1977 Biochemistry).

(2). The AGPC method has been widely considered the gold standard in RNA extraction (Chomczynski & Sacchi 1987, cited by more than 70,000 times). Our study showed that crosslinking-induced higher hydrophobicity, a fundamental new property that does not depend on specific crosslinkers (see **Supplementary Figs. 4-5**). Our work revealed an important caveat in the applications of the classical AGPC method (e.g. TRIzol) that should be considered in any future applications of nucleic acid crosslinkers. In another study, it was shown that structured microRNAs with low GC content are selectively lost during TRIzol extraction from a small number of cells, further demonstrating the potential problems associated with typical TRIzol extractions (Kim et al. 2012, Mol Cell, PMID: 22749402).

It is important to note that the TNA method we developed is absolutely essential for full recovery of crosslinked intact RNA and subsequent targeted enrichment of specific subsets of transcripts, which has not been possible before. We demonstrated that all previous methods either caused RNA fragmentation before enrichment (PARIS, Lu et al. 2016 Cell), or resulted in severe and selective loss of longer crosslinked RNAs (e.g. SPLASH, LIGR-seq and COMRADES which uses TRIzol or RNeasy, see **Supplementary Fig. 6**).

(3). UV is widely used in molecular biology and photochemical damages pose serious limits on the detection of nucleic acids. In this manuscript, our deep mechanistic analysis led to new insights into the mechanisms and solutions to mitigate the damages. For example, we showed for the first time that antioxidants and radical scavengers previously thought to prevent PUVA-induced oxidative damages actually block crosslinking because crosslinking and oxidative damages are mediated by the same energetic precursors. As a result, these compounds should NOT be used to ameliorate the side-effects of crosslinking (**Supplementary Fig. 14**). UVC-induced photochemical damages of RNA were recognized since at least the 1950s, and we showed for the first time that such damages can be prevented using singlet quenchers when used at high enough concentrations (**Supplementary Figs. 10-11**). Our extensive screening of conditions revealed that promotion of reverse transcriptase dynamics can efficiently bypass RNA damages (**Supplementary Fig. 15**). Together, these new insights and methods will be widely applicable to any RNA research that involves damaged RNAs.

(4). As requested by the reviewer, we have now performed a direct comparison between PARIS1 and PARIS2 results (see **response to reviewer 1, points 1-2**, revised **Supplementary Fig. 2**). With the dramatic >4000-fold improvement in efficiency, there is no reduction in accuracy or specificity. We do acknowledge that the uridine preference of psoralens remains one of the most serious limitations, and the amotosalen introduced in this manuscript still did not solve this problem. We are currently developing new chemical crosslinkers that has no bias towards the bases. Despite this limit, we believe that the broad applicability of the new methods in RNA research, >4000-fold increased efficiency and enabling targeted enrichment (mRNAs, snoRNAs, viral RNA genomes) represent “meaningful enhanced results”.

2. The authors went on to demonstrate three applications of PARIS2, which yielded a large amount of data of RNA structures and interactions. However, the study largely fell short of validating these interactions with alternative methods or functional experiments. Therefore, it would be difficult to access the degree of false positive in the results.

For example, it would be helpful for the authors to compare the mRNA-ribosome binding data with previously described Riboseq data to cross validate the results. For another example, it would be helpful to perform mutation experiment of the virus genome sequence to validate the functional importance of detected interaction. These experiments would be very important for validation of the profiling results as well as the methodology.

We thank the reviewer for this great suggestion. Here we describe our original experiments, analysis and revisions to address the concerns. **(1)**. We now present a more detailed analysis of accuracy and specificity, using the ribosomal RNAs and snRNAs, which have been well studied before (new panels in **Supplementary Fig. 2**, see response to reviewer 1, points 1-2). The accuracy, and specificity of PARIS2 is similar to PARIS1, which was already extensively tested in our earlier paper (Lu et al. 2016). **(2)**. For the analysis of mRNA-rRNA interaction, we now present a more detailed comparison, using more PARIS2 data and published Riboseq data (new panels in **Supplementary Fig. 19**, see response to reviewer 1, point 6). **(3)**. We have performed further validations on the U8-U13 interaction and the mutation-induced destabilization of U8 structures and dimerization (new panels in **Supplementary Fig. 20-21**, see response to reviewer 1, points 7-8). **(4)**. The newly described structures in the viral RNAs are extensively validated using multiple approaches, including cross-validation in multiple different experimental conditions and phylogenetics support (see response to reviewer 1, point 11). **(5)**. We also performed mutational analysis on a newly discovered alternative structure in the IRES element and showed that it is required for optimal IRES activity (**Supplementary Fig. 32**, see response to reviewer 1, point 12).

3. On the format, the writing and organization of the manuscript made it a bit difficult to read. Some of the technical descriptions in the main text could be transferred to supplementary materials to make the main text more friendly to general readership. The figures, especially Figure 5 and 6 are somewhat too crowded. Some trimming or reorganization would be helpful.

We apologize for the problems in data presentation. As described above (response to point 1), the technology development in this manuscript spanned multiple diverse disciplines, and we carried out extensive mechanistic studies into photochemistry, RNA chemistry and enzymology. The new insights are invaluable for the field, which is why we felt it necessary to report them in detail. In fact, we were still unable to describe them completely in the main text due to space limit. To help readers understand and apply these new techniques, we wrote 5 lengthy **Supplementary Notes**, each organized like a “mini-paper” to describe the background of each problem, our mechanistic studies, comprehensive screening of chemicals and enzymes, and the final optimal conditions for PARIS2 (**Supplementary Figs. 1-15**). For **Figures 5 and 6**, we have reorganized the panels to facilitate reading.

REVIEWER COMMENTS

Reviewer #1 (Remarks to the Author):

The manuscript from Zhang M et al has been substantially revised. However, the revision does not address the most important question of how this protocol is an improvement over PARIS. This reviewer understands that with the change in crosslinker, and extraction strategies etc, the sensitivity of PARIS2 is supposed to be 4000X than of PARIS. However, the authors did not convincingly show an example whereby one would not be able to capture something using PARIS and would be able to do so accurately using PARIS2. Without such a convincing example, it is difficult for users of the technology to determine when they should be using PARIS2 instead of PARIS.

Here are the major comments:

- 1) The authors showed many examples of interactions that are captured by PARIS2 in their manuscript, and that additional crosslinking did not result in poorer data quality. The authors should show convincingly a situation, whereby sequenced to the same depth as PARIS, PARIS2 is able to capture an important interaction that PARIS cannot, probably on a poorly expressed transcript etc.
- 2) With regards to the + and - strand structure probing of enteroviruses, did the authors use actinomycin during their RT? Could the detection of the -ve strand be an artifact?
- 3) In the example whereby the authors disrupted the IV domain of the 5'UTR of the enterovirus, the authors disrupted the IV domain as well as performed point mutations on the alternative 3a structure. How is the IV domain disrupted? Which bases are mutated?
- 4) This reviewer does not understand why compensatory mutations cannot be made on the IV or 3a structure given that the region that the authors are probing is in the 5'UTR and is not constrained by the coding potential of the virus.
- 5) With regards to using PARIS2 as an estimate for translation efficiency- what are the limitations of using PARIS2 to estimate TE. Is the method only accurate for abundant transcripts? Is there a lower threshold below which one would not trust the results?

Minor comment

- 1) The authors use the word "re-invented" in the abstract. Please use the word "improved".
- 2) There are numerous strong claims in this manuscript that should be toned down.

Reviewer #2 (Remarks to the Author):

I thank the authors for their efforts to improve the initial submission of this manuscript.

I have some remaining comments, but overall, I believe this version is appropriate for publication.

1) I am impressed and very supportive of the effort put into developing the experiments and further understanding all the technological advances and limitations. Many of which had to be overcome to optimize the PARIS method. I applaud the authors in their systematic and thorough analyses presented herein.

2) The authors state in their rebuttal that " With the dramatic >4000-fold improvement in efficiency, there is no reduction in accuracy or specificity." This in conjunction with Supplementary Figure 2 further supports my initial concern -- that there is not a significant difference between the probes and that data they provide. This is a practical issue for usage by the field because psoralen can be purchased and used off the shelf, while the new probe which is supposed to provide a substantial improvement does not seem to be the case. I agree that the crosslinking efficiency is improved, but the added crosslinking sites do not seem to be significant enough to alter the analysis of dsRNA structures or RNA-RNA interactions to yield novel insight that could not be obtained with psoralen.

3) for the analysis of Supplementary Figure 19, the authors put forth a substantial amount of effort to demonstrate that PARIS signal is enriched in the coding region of mRNAs. The logic behind this is that rRNA-mRNA interactions are driving crosslinking. However, there is no detailed analysis of the chimeric reads between mRNA and rRNA itself. Instead it is more of an analysis of crosslinked mRNA. This does not represent mRNA-rRNA interactions, as the authors are claiming. In fact, one would predict, based on ribosome helicase activity that the mRNA would have LESS crosslinking of mRNA-mRNA interactions in coding regions and not in UTRs. This should be developed more in future manuscripts.

Reviewer #3 (Remarks to the Author):

The authors have made a great effort to improve the manuscript. However, I agree with Reviewer 1, this study would benefit from being split into 2-3 better developed individual manuscripts. The added supplementary Notes are too lengthy in my opinion, which could be developed into a review or a protocol paper in the future. With regards to my specific comments in my last review, I am not very convinced that the revision has addressed all my concerns.

Regarding the comparison between PARIS2 and PARIS, Supplementary Fig 2i & 2j show that the two methods worked very similarly, so I am confused about the degree of improvement. The comparison with Riboseq data is convincing, but again, I am not sure why people would choose PARIS2 method over Riboseq for this application as Riboseq is quite robust now.

Regarding the validation of the viral RNA structure, I am afraid that the reporter system the authors used might not reflect the transcription of the virus genome. Ideally, the mutations should be generated in the genome of recombinant virus to measure their effect on viral replication.

On the format, the manuscript is still too crowded for both the text and the figures. The manuscript would benefit from using larger font, more spacing in the text. The figures could be enlarged to facilitate reviewing.

REVIEWER COMMENTS

Reviewer #1 (Remarks to the Author):

The manuscript from Zhang M et al has been substantially revised. However, the revision does not address the most important question of how this protocol is an improvement over PARIS. This reviewer understands that with the change in crosslinker, and extraction strategies etc, the sensitivity of PARIS2 is supposed to be 4000X than of PARIS. However, the authors did not convincingly show an example whereby one would not be able to capture something using PARIS and would be able to do so accurately using PARIS2. Without such a convincing example, it is difficult for users of the technology to determine when they should be using PARIS2 instead of PARIS.

We appreciate the reviewer's recognition of the extremely enhanced sensitivity of PARIS2. On one hand, we acknowledge that the fundamental limitation of psoralen crosslinking bias remains, therefore, PARIS2 captures similar duplexes as the original version. We added a note in the discussion regarding this limitation "New crosslinkers that overcome the psoralen (both AMT and amotosalen) bias toward uridines will enable the analysis of previously uncrosslinkable duplexes and further improve crosslink-ligation based methods"

On the other hand, we want to point out that, in addition to the higher sensitivity, the ability to enrich specific RNAs using antisense probes is essential for many applications. This is only possible due to our new TNA method. Indeed, as mentioned in the previous revision, most requests for help and collaborations that we received in the past few years were on specific disease-associated RNAs, which are often expressed at much lower levels compared to abundant noncoding RNAs and housekeeping gene mRNAs. More importantly, many animal and clinical studies only yield limited amounts of samples, which necessitate more sensitive analytical methods. There is no reason why anyone does not want to use a more sensitive method when it does not require more efforts. Therefore, PARIS2 will certainly replace the first-generation PARIS method.

In the supplementary protocol, we have added another note on the flexibility of applying the improvements in PARIS2: "Due to the modular design of PARIS2, users can apply some of the improvements while not using others. For example, if amotosalen is not available to the user, AMT can be used, in combination with all other improvements. In this case, other improvements will remain effective."

Here are the major comments:

1) The authors showed many examples of interactions that are captured by PARIS2 in their manuscript, and that additional crosslinking did not result in poorer data quality. The authors should show convincingly a situation, whereby sequenced to the same depth as PARIS, PARIS2 is able to capture an important interaction that PARIS cannot, probably on a poorly expressed transcript etc.

As mentioned above, the psoralen bias toward uridines remains a major limitation, which requires new chemistry that is beyond the scope of this study. As a result, PARIS2 does not capture duplexes that cannot be captured by PARIS. Our last revision showed that when looking at the same sequencing depth, PARIS2 performs essentially the same as PARIS. The value of PARIS2 lies in the extremely high sensitivity and the ability to perform targeted analysis of low abundance RNAs in smaller amounts of starting materials (again, see details in **Supplementary Tables**). This advantage has been clearly demonstrated in the applications to mRNAs, snoRNAs and viral RNAs.

2) With regards to the + and - strand structure probing of enteroviruses, did the authors use actinomycin during their RT? Could the detection of the -ve strand be an artifact?

We did not use actinomycin in the reverse transcription experiments. In order to rule out the possibility of spurious second strand synthesis during reverse transcription, we already identified reads mappable to the (+) and (-) strands of ribosomal RNAs. Compared to the EV-D68 viral RNA, where the (-) strand is about 0.4-3.7% of the (+) strand, (-) strand reads on rRNA (in the same library and dataset) were barely detectable (**Supplementary Fig. 29d**, copied here to facilitate review). Therefore, we conclude that the spurious second strand synthesis can be ignored and the structure data on the (-) strand are not artifacts. The relative abundance of (+) and (-) strand RNA is also consistent with previous rigorous analysis of Picornavirus replication intermediates (Novak & Kirkegaard 1991, J Virology).

In addition, the following new observations are also consistent with the idea that the (-) strand data are not artifacts. (1). The density of double stranded regions captured by PARIS2 is highly enriched in crosslinkable sites, i.e. UA motif and A/U rich sequences (**Supplementary Fig. 29i-j**). (2). The 5' end ~700nts has significantly lower coverage on the

negative strand compared to the rest of the genome (**Fig. 5p**). This phenomenon cannot be explained by the (-) strand being reverse transcription artifacts, since these artifacts, if true, should result in uniform coverage along the genome.

3) In the example whereby the authors disrupted the IV domain of the 5'UTR of the enterovirus, the authors disrupted the IV domain as well as performed point mutations on the alternative 3a structure. How is the IV domain disrupted? Which bases are mutated?

We apologize for the confusion. We have now updated the diagrams in **Supplementary Fig. 32b,d**, where the mutations are indicated by larger arrow heads. In addition, the details of the mutations, including sequences and minimal free energy of the mutant structures, are listed in **Supplementary Methods** and **Supplementary Table 5**.

4) This reviewer does not understand why compensatory mutations cannot be made on the IV or 3a structure given that the region that the authors are probing is in the 5'UTR and is not constrained by the coding potential of the virus.

The reviewer is correct in pointing out that the numbers of constraints determine the difficulty in designing compensatory mutations. In the case of the domain IV and 3a, these two conformations are alternative to each other, where compensatory mutations for one conformation will disrupt the other, making it difficult to interpret the consequences. As a result, we were not able to introduce compensatory mutations without unintended side effects.

5) With regards to using PARIS2 as an estimate for translation efficiency- what are the limitations of using PARIS2 to estimate TE. Is the method only accurate for abundant transcripts? Is there a lower threshold below which one would not trust the results?

The limitation of PARIS2 in measuring translation efficiency lies in its lower sensitivity compared to Riboseq – only ~10% of PARIS2 reads are gapped due to the low efficiency of proximity ligation. As with any sequencing-based quantification, accuracy depends on sequencing depth (e.g. SEQC/MAQC-III Consortium 2014, <https://www.nature.com/articles/nbt.2957>), the higher abundance, the higher accuracy. As usual, it is difficult to determine specific threshold or cutoff for reliable vs. non-reliable results. We have now provided RPKM values comparing PARIS2 and Riboseq in **Supplementary Fig. 19j and k** in the Source Data (**sfig19j** subfolder). We have toned down and clarified this point in the main text: *“PARIS2 does not replace ribosome profiling, due to its lower efficiency (~10% reads are gapped), however, the ability to capture translation status is a bonus during the analysis of mRNA structures and interactions.”*

Minor comment

- 1) The authors use the word “re-invented” in the abstract. Please use the word “improved”.
- 2) There are numerous strong claims in this manuscript that should be toned down.

We have revised the text according to the reviewer’s comments. We have toned down the claim on rRNA-mRNA interactions profiling: *“PARIS2 does not replace ribosome profiling, due to its lower efficiency (~10% reads are gapped), however, the ability to capture translation status is a bonus during the analysis of mRNA structures and interactions.”* We also included an additional note on the psoralen bias: *“New crosslinkers that overcome the psoralen (both AMT and amotosalen) bias toward uridines will enable the analysis of previously uncrosslinkable duplexes and further improve crosslink-ligation based methods.”*

Reviewer #2 (Remarks to the Author):

I thank the authors for their efforts to improve the initial submission of this manuscript. I have some remaining comments, but overall, I believe this version is appropriate for publication.

We sincerely thank the reviewer for the support of publication of our manuscript.

1) I am impressed and very supportive of the effort put into developing the experiments and further understanding all the technological advances and limitations. Many of which had to be overcome to optimize the PARIS method. I applaud the authors in their systematic and thorough analyses presented herein.

We really appreciate the reviewer’s recognition of the technological innovations in this manuscript. Indeed, I started developing PARIS as a “rogue” side project while I was finishing my PhD in the Matera lab at UNC Chapel Hill in 2013, then continued through my postdoctoral training in the Chang lab at Stanford, and now in my own lab. Some of the technological problems were noticed since almost 8 years ago (e.g. loss of crosslinked RNA in TRIZOL extraction, and extensive UV damages). We are really proud to be able to finally understand these fundamental problems and push the technology to its limit.

2) The authors state in their rebuttal that " With the dramatic >4000-fold improvement in efficiency, there is no reduction in accuracy or specificity." This in conjunction with Supplementary Figure 2 further supports my initial concern -- that there is not a significant difference between the probes and that data they provide. This is a practical issue for usage by the field because psoralen can be purchased and used off the shelf, while the new probe which is supposed to provide a substantial improvement does not seem to be the case. I agree that the crosslinking efficiency is improved, but the added crosslinking sites do not seem to be significant enough to alter the analysis of dsRNA structures or RNA-RNA interactions to yield novel insight that could not be obtained with psoralen.

We acknowledge that amotosalen does not solve the problem of biased uridine crosslinking. The value in PARIS2 lies in its extremely high efficiency and the ability to analyze specific transcripts using antisense oligo mediated enrichment. Together, these improvements, even in the absence of amotosalen, made it possible to apply PARIS2 to limited amounts of animal and clinical samples and/or low abundance transcripts, which was difficult using previous methods. Therefore, PARIS2 will find broader use in biology than the original version. To demonstrate this point, we showed applications on low abundance snoRNAs and viral genome RNAs (Fig. 4-6).

The reviewer raised a great point that the commercial availability of amotosalen will limit its applications. We believe this is only a minor and temporary issue. First, with any new technology, widespread adoption takes time. With our demonstration of its superior solubility and crosslinking efficiency, we believe that chemical vendors will begin supplying amotosalen for research purposes. We have also been sending it to others upon request since our initial posting of the manuscript to Biorxiv.

In addition, we would like to point out that all the improvements can be applied individually or in combinations. The ~7-fold higher efficiency of amotosalen is only part of the >4000-fold higher sensitivity. Even in the absence of amotosalen, one can still apply all the other improvements described in the manuscript and achieve several hundred folds increased sensitivity. For example, we demonstrated that from the same amount of crosslinked RNA, the improved library preparation steps alone resulted in ~76-fold higher sensitivity (Supplementary Tables 1-2). To help users, we added another note to the **Supplementary Protocol**: "Due to the modular design of PARIS2, users can apply some of the improvements while not using others. For example, if amotosalen is not available to the user, AMT can be used, in combination with all other improvements. In this case, other improvements will remain effective."

3) for the analysis of Supplementary Figure 19, the authors put forth a substantial amount of effort to demonstrate that PARIS signal is enriched in the coding region of mRNAs. The logic behind this is that rRNA-mRNA interactions are driving crosslinking. However, there is no detailed analysis of the chimeric reads between mRNA and rRNA itself. Instead it is more of an analysis of crosslinked mRNA. This does not represent mRNA-rRNA interactions, as the authors are claiming. In fact, one would predict, based on ribosome helicase activity that the mRNA would have LESS crosslinking of mRNA-mRNA interactions in coding regions and not in UTRs. This should be developed more in future manuscripts.

c

We apologize for the confusing description, which led to misunderstanding of the analysis method. We have now added diagrams in Fig. 4 to describe the method (see screenshot on the right). For Fig. 4c, we extracted gapped reads with one arm mapped to hs45S and the other arm mapped to other RNAs. Then we calculated the coverage on hs45S for reads with one arm mapped to one of the three categories, mRNA, hs12S (mitochondrial small subunit ribosomal RNA) and hs16S (mitochondrial large subunit ribosomal RNA). Therefore, the peaks on the 45S represent direct interactions with mRNAs. In particular, 2 major regions in hs45S, h18 and h26, contact mRNAs. As negative controls, no specific direct interactions were observed between cytoplasmic and mitochondrial ribosomal RNAs.

Similarly in Fig. 4d, we extracted gapped reads with one arm mapped to hs45S h18 or h26 and the other arm mapped to mRNAs, and then we calculated the coverage of reads mapped to a meta mRNA. Therefore, the coverage on mRNA represent direct interactions between h18 or h26 with mRNAs. More importantly, these results are consistent with previous work that showed direct rRNA-mRNA crosslinking using 254nm UV (Pisarev et al. 2008 EMBO J).

d

Reviewer #3 (Remarks to the Author):

The authors have made a great effort to improve the manuscript. However, I agree with Reviewer 1, this study would benefit from being split into 2-3 better developed individual manuscripts. The added supplementary Notes are too

lengthy in my opinion, which could be developed into a review or a protocol paper in the future. With regards to my specific comments in my last review, I am not very convinced that the revision has addressed all my concerns.

As seen in earlier critiques, reviewers requested additional evidence to demonstrate the broad applicability of PARIS2 and its advantage over the original method (e.g. reviewer 1, points 3 and 5, reviewer 2 point 2 in the first round of review.). Our initial inclusion of extensive validations and multiple diverse applications was exactly meant for this purpose. We would certainly be happy to take out some of the sections for future papers if all reviewers agree, but we are afraid this will weaken the claims.

We appreciate the recommendation of publishing the **Supplementary Notes** separately, which certainly will be great for us just to have more papers. However, in our many years of experience in developing and applying new technologies, the biggest challenge that we have encountered is often in finding detailed description of earlier methods in order to design and implement improvements (as evidenced in the large numbers of references that we had to find and include in the **Supplementary Notes**, many of which date back to the 1960s). We believe this is also the unfortunate experience of many others. Because of this, we strived to provide a complete description of the essential background and our optimizations in the same manuscript. Our hope is that these Supplementary Notes will serve as the definitive guideline and reference for the entire field. In addition, we have toned down and clarified some of the claims as recommended (see below).

Regarding the comparison between PARIS2 and PARIS, Supplementary Fig 2i & 2j show that the two methods worked very similarly, so I am confused about the degree of improvement. The comparison with Riboseq data is convincing, but again, I am not sure why people would choose PARIS2 method over Riboseq for this application as Riboseq is quite robust now.

Psoralens have the inherent bias towards uridines; as a result, the two versions produce very similar results at the same level of sequencing coverage. We acknowledge that amotosalen does not overcome this bias. However, the advantage of PARIS2 lies in the extremely high sensitivity and the ability to analyze low abundance RNAs using antisense-mediated enrichment. These improvements are critical for many biological applications, as demonstrated in the examples described in the manuscript.

We acknowledge that PARIS2 does not replace Riboseq given its lower efficiency (~10%) in the production of gapped reads. We have clarified this point in the main text: "*PARIS2 does not replace ribosome profiling due to its lower efficiency (~10% reads are gapped), however, the ability to capture translation status is a bonus during the analysis of mRNA structures and interactions.*"

Regarding the validation of the viral RNA structure, I am afraid that the reporter system the authors used might not reflect the transcription of the virus genome. Ideally, the mutations should be generated in the genome of recombinant virus to measure their effect on viral replication.

The IRES activity reporter was not designed to test the effect of RNA structural alterations on genome transcription or replication. As such, we did not make any claim about these consequences. We only used these bicistronic reporters to test their effects on translation, which is a standard approach in the field (e.g. Weingarten-Gabbay et al. 2016, <https://science.sciencemag.org/content/351/6270/aad4939>).

On the format, the manuscript is still too crowded for both the text and the figures. The manuscript would benefit from using larger font, more spacing in the text. The figures could be enlarged to facilitate reviewing.

We apologize for the formatting issues. We have now revised the figures to use larger font and more space to facilitate reading.

REVIEWER COMMENTS

Reviewer #1 (Remarks to the Author):

The authors have addressed most of my concerns except for the one that compares PARIS to PARIS2- which is probably the most important concern for this paper. This is important because readers need to know when to use PARIS2 over PARIS and whether they need to switch to PARIS2 if they have been using PARIS. However the authors have not convincingly showed how PARIS2 is better than PARIS. The authors claim that PARIS2 is 4000X more sensitive than PARIS. How does this translation to the actual experimental protocol? Does that mean that PARIS2 can start with a much small amount of starting material as compared to PARIS? Does that mean that PARIS2 can detect RNA-RNA interactions in less abundant transcripts? Does that mean that PARIS2 can identify the same crosslinks with lower amount of sequencing? The authors need to convincingly show one example of improvement over PARIS for PARIS2.

Reviewer #2 (Remarks to the Author):

The authors have addressed all of my concerns. I believe this manuscript is now at the level of publication.

Reviewer #3 (Remarks to the Author):

The authors have addressed my concerns. I recommend the publication of this paper.

We appreciate the reviewers' constructive comments. Please see response to Reviewer 1 below.

The authors have addressed most of my concerns except for the one that compares PARIS to PARIS2- which is probably the most important concern for this paper. This is important because readers need to know when to use PARIS2 over PARIS and whether they need to switch to PARIS2 if they have been using PARIS. However, the authors have not convincingly showed how PARIS2 is better than PARIS. The authors claim that PARIS2 is 4000X more sensitive than PARIS. The authors need to convincingly show one example of improvement over PARIS for PARIS2.

The reviewer raised a valid concern on the transition of PARIS to PARIS2. In general, switching methods in the middle of a project is not a good idea, even though we showed that the structures captured by PARIS and PARIS2 are comparable at the same sequencing depth (**Supplementary Fig. 2**). However, if one were to start a new project, PARIS2 is definitely better due to its much higher efficiency and the ability to analyze specific subsets of RNAs. We want to emphasize that we have already provided not just one example of improvement, but many, from the mechanistic studies of photochemistry and enzymology, to the actual library preparations and the applications to specific biological problems. Answers to the reviewer's specific questions are as follows.

How does this translation to the actual experimental protocol?

In our initial submission, we already provided extensive quantitative comparisons between the two methods (see **Supplementary Tables 1-3**), and a step-by-step protocol for PARIS2. We now copy **Supplementary Tables 1-2** here to further explain the improvements in detail. In general, almost every step has been significantly improved (with the exception of proximity ligation).

(1) In sample preparation (shown below), the improvements in the yield of crosslinked RNA fragments from the same amount of starting materials means that we can use less starting materials to get the same amount of cDNA library.

Sample preparation	Comparison	Average Ratio	Related figures
Amotosalen crosslinking • 0.5 mg/ml amotosalen • 2.0 mg/ml amotosalen • 5.0 mg/ml amotosalen	0.5 AMT	3.42–6.95 • 3.42 • 5.13 • 7.95	Supplementary Fig. 2d
TNA method • 0.5 mg/ml amotosalen • 5.0 mg/ml amotosalen	TRizol	2.13–6.48 • 2.13 • 6.48	Fig. 2k Supplementary Fig. 3 and 6
DD2D gel purification	ND2D	1.54	Supplementary Fig. 8
Total improvement ratio • 0.5 mg/ml amotosalen • 5.0 mg/ml amotosalen	PARIS1	11.21–69.35 • 11.21 • 69.35	

(2). In library preparation (shown below), improved photochemistry and enzymology resulted in ~76 fold more cDNA from the same amount of crosslinked RNA fragments. This means that more cDNA library is available for sequencing, making it possible to sequence deeper to detect structures and interactions on less abundant transcripts.

Library preparation (after 2D gel)	Comparison	Average Ratio	Related figures
Adapter ligation • SLRNA2 • SLRNA8	Standard	2.45–3.91 • 2.45 • 3.91	Supplementary Fig. 12
UVC protection (AO) • based on GAPDH (70 bp) • based on ACTB (184 bp)	no AO	6.51 - 9.90 • 6.51 • 9.90	Fig. 2m-n Supplementary Fig. 10
PUVA bypass (SSIV)	SSIII	3.32	Fig. 2o
PUVA bypass (SSIV with Mn ²⁺ buffer) • based on SONRD118 (96 bp) • based on SNORD13 (63 bp) • based on ACTB (184 bp)	Mg ²⁺ buffer	2.48 - 23 • 2.48 • 2.62 • 23.03	Fig. 2p Supplementary Fig. 15
Total library yield	PARIS1	76	

(3). The numbers on actual library preparations are in **Supplementary Table 2**, total library yield (copied below). In 6 trials during the span of several months, all starting from 50ng crosslinked RNA fragments, we were able to get an average ~76-fold higher cDNA library yield. These experiments provided further concrete examples on the improvements of PARIS2.

Total library yield		
Library yield (nmol)	PARIS1	PARIS2
Rep1	0.38	60.19
Rep2	0.71	51.86
Rep3	1.59	93.17
Rep4	0.49	42.86
Rep5	NA	58.85
Rep6	NA	53.27
Average yield	0.79	60.03
Ratio (to PARIS1)		75.75

Note: Library products yield from 50 ng of crosslinked RNA.

Does that mean that PARIS2 can start with a much small amount of starting material as compared to PARIS?

Yes. As shown in the table above (**Supplementary Table 1, sample preparation**), PARIS2 (5mg/ml amotosalen crosslinked) produces 69-fold more crosslinked RNA fragments than PARIS1 (0.5mg/ml AMT crosslinked) from the same number of cells. This means that we can get the same amount of crosslinked RNA fragments using 1/69 amount of cells. Although we did not test this amount (1/69) specifically, we have been using less starting material for PARIS2 and obtained higher quality libraries. In **Supplementary Table 2 Total library yield** (shown above), we presented actual data on many library preparation trials, which showed the consistent higher yield. Again, this improvement (69 fold higher yield of crosslinked RNA) is only part of the overall >4000 fold higher efficiency.

Does that mean that PARIS2 can detect RNA-RNA interactions in less abundant transcripts?

Yes. In theory, the ability to detect a specific crosslinking event, either intramolecular duplexes or intermolecular interactions depends three factors: (1) relative abundance of the transcript in the pool, (2) the frequency of gapped reads (~10% proximity ligation efficiency in our most advanced protocol), and (3) the total sequencing depth. More specifically:

(1). In PARIS2, the newly invented TNA method allows quantitative isolation of intact crosslinked RNA from cells for the first time, making it possible to perform antisense-mediated enrichment and therefore increasing the “relative abundance of the transcript in the pool” (**factor 1**). We have demonstrated this point using three distinct examples, mRNAs, snoRNAs and viral genome RNAs (**Figs. 4-6**). For example, in **Supplementary Fig. 16c**, we showed that mRNAs are enriched to ~80% from both human 293T cells and mouse brain, whereas they normally represent <5% in total RNA. In **Supplementary Fig. 22i**, we showed that viral RNAs are enriched by ~100-fold compared to cellular RNAs (e.g. beta-ACTIN mRNA and 18S rRNA). The details of the newly structures and interactions are described extensively in the manuscript, so we will not repeat them here.

(2). We have not been able to improve proximity ligation efficiency (**factor 2**). In fact, we have not seen any improvement in the entire field, despite more than 10 years of effort across many labs.

(3). The total sequencing depth (**factor 3**) depends on how much cDNA library is produced from the starting material, and how much money one is willing to spend on sequencing. The dramatically improved efficiency in PARIS2 means that one can produce much more cDNA library from the same amount of RNA (**Supplementary Table 1**), therefore making it possible to detect less abundant transcripts, as long as one wants to sequence more.

Taken together, we proved beyond doubt that PARIS2 can detect RNA-RNA interactions in less abundant transcripts.

Does that mean that PARIS2 can identify the same crosslinks with lower amount of sequencing?

This is theoretically impossible without improving proximity ligation, because the amount of crosslinks is proportional to the amount of sequencing. In any sequencing experiment, each read provides one quantal unit of information. In the case of PARIS and related methods that determine the physical interaction between two pieces of RNA, each gapped read provides one quantal unit of information on the base-pairing interaction. Since in PARIS and related methods, gapped reads only represent a small fraction (up to ~10% in our data), the ONLY factor that determines the amount of useful information for each read (or a specific number of reads) is the proximity ligation efficiency. As

described in the manuscript, there has been little improvement despite over 10 years of effort in the field by many laboratories (testing various types of ligases, linkers, conditions, etc.). For example, if we sequence 10 reads and one of them is a gapped read reporting a RNA duplex (10% efficiency), we cannot sequence fewer reads to get the 1 gapped read, without improving proximity ligation.